# Post-ER Stress Biogenesis of Golgi Is Governed by Giantin

**DOI:** 10.3390/cells8121631

**Published:** 2019-12-13

**Authors:** Cole P. Frisbie, Alexander Y. Lushnikov, Alexey V. Krasnoslobodtsev, Jean-Jack M. Riethoven, Jennifer L. Clarke, Elena I. Stepchenkova, Armen Petrosyan

**Affiliations:** 1Department of Biochemistry and Molecular Biology, University of Nebraska Medical Center, Omaha, NE 68198-5870, USA; cole.frisbie@unmc.edu; 2Nanoimaging Core Facility, University of Nebraska Medical Center, Omaha, NE 68198-6025, USA; alushnikov@unmc.edu (A.Y.L.); akrasnos@unomaha.edu (A.V.K.); 3Department of Physics, University of Nebraska-Omaha, Omaha, NE 68182-0266, USA; 4Center for Biotechnology, University of Nebraska-Lincoln, Lincoln, NE 68588-0665, USA; jeanjack@unl.edu; 5The Nebraska Center for Integrated Biomolecular Communication, University of Nebraska-Lincoln, Lincoln, NE 68588-0304, USA; jclarke3@unl.edu; 6Department of Statistics, University of Nebraska-Lincoln, Lincoln, NE 68583-0963, USA; 7Vavilov Institute of General Genetics, Saint-Petersburg Branch, Russian Academy of Sciences, Saint-Petersburg 199034, Russia; stepchenkova@gmail.com; 8Department of Genetics, Saint-Petersburg State University, Saint-Petersburg 199034, Russia; 9The Fred and Pamela Buffett Cancer Center, Omaha, NE 68198-5870, USA

**Keywords:** Brefeldin A, Golgi biogenesis, giantin, Rab6a, GRASP65

## Abstract

Background: The Golgi apparatus undergoes disorganization in response to stress, but it is able to restore compact and perinuclear structure under recovery. This self-organization mechanism is significant for cellular homeostasis, but remains mostly elusive, as does the role of giantin, the largest Golgi matrix dimeric protein. Methods: In HeLa and different prostate cancer cells, we used the model of cellular stress induced by Brefeldin A (BFA). The conformational structure of giantin was assessed by proximity ligation assay and atomic force microscopy. The post-BFA distribution of Golgi resident enzymes was examined by 3D SIM high-resolution microscopy. Results: We detected that giantin is rather flexible than an extended coiled-coil dimer and BFA-induced Golgi disassembly was associated with giantin monomerization. A fusion of the nascent Golgi membranes after BFA washout is forced by giantin re-dimerization via disulfide bond in its luminal domain and assisted by Rab6a GTPase. GM130-GRASP65-dependent enzymes are able to reach the nascent Golgi membranes, while giantin-sensitive enzymes appeared at the Golgi after its complete recovery via direct interaction of their cytoplasmic tail with N-terminus of giantin. Conclusion: Post-stress recovery of Golgi is conducted by giantin dimer and Golgi proteins refill membranes according to their docking affiliation rather than their intra-Golgi location.

## 1. Introduction

The Golgi complex in one of the largest cellular organelles, which is increasingly viewed as the main coordinator of different trafficking pathways and molecular events that play a grand role in cellular homeostasis [1,2]. For years, researchers have obsessed over the molecular mechanisms that contribute to Golgi function and unique ribbon-like architecture. The emerging consensus is that the sequential distribution of enzymes in the interconnected stacks of flat cisternae provides the appropriate modifications to the cargo [3,4]. This is the main paradigm of secretory transport through the Golgi, regardless of whether it is postulated by the cisternal or vesicular model [5,6]. Therefore, the intact and perinuclear position of Golgi is ideal for the complete processing of proteins. However, in response to different intracellular perturbations, including inhibition of endoplasmic reticulum (ER) to Golgi trafficking and subsequent ER stress, as well as the treatment with or consumption of various chemicals and alcohol, Golgi undergoes disorganization, characterized by unstacking its membranes and abnormal glycosylation [7,8,9,10,11,12,13]. Nevertheless, Golgi has a phenomenal self-organizing mechanism [14,15], and, under drug- or stress-free conditions, Golgi is able to return to its classical ribbon-like structure and perinuclear positioning [16,17,18,19]. This fact is of particular note because, despite over 100 years of Golgi research, scientists are still at a loss to explain the mechanisms of Golgi’s ability to maintain and recover harmony in such complicated structure. 

It has now been more than thirty years since fungal metabolite Brefeldin A (BFA) was employed as a model for the effective blockade of ER-to-Golgi transportation and rapid Golgi disorganization [20]. BFA-induced Golgi disassembly is reversed upon drug washout (WO) [16,17], and BFA acts as an uncompetitive inhibitor that binds to the Arf1–GDP–GEF complex, thus blocking the activation of ARF1 and subsequent COPI coat assembly [21,22,23]. Indeed, loss of COPI vesicles from the Golgi is one of the earliest responses to BFA treatment [24]. However, knockdown (KD) of COPI coatomer proteins, including β-COP, despite Golgi disorganization, does not mimic the crucial effect of BFA, i.e., collapse of Golgi membranes and their absorption into the ER [25,26,27,28,29]. This suggests that BFA is the drug having the potential to perform multiple tasks, especially in that it may have an effect on the structure of Golgi scaffold proteins.

The integrity of the Golgi requires structural cooperation of its matrix (golgins) and residential proteins [30,31,32]; in turn, Golgi compact organization is likely to be a paramount condition for the successful trafficking of some critical enzymes to the Golgi [33,34]. While enzymes are responsible for the processing of proteins, golgins serve as the guardians of Golgi’s monolithic architecture and docking sites for many Golgi targeting vesicles [35,36,37,38,39]. The function of golgins is coordinated by Golgi reassembly stacking proteins (GRASPs) [40,41,42]. We recently identified that Golgi glycosyltransferases employ two different recruiting sites at the Golgi: giantin and the complex of GM130 and GRASP65 [43]. Giantin is the largest (376 kDa) Golgi matrix protein. It has a long (≥350 kDa) cytoplasmic N-terminal region followed by one-pass trans-membrane domain and short Golgi lumenal C-terminal domain which stabilizes dimeric giantin via a disulfide bond [44,45]. It has been suggested that giantin could be essential in the cross-bridge structures that maintain Golgi morphology and it is required for efficient SNARE-mediated fusion [44,46]; however, this has never been directly tested. A strong argument in defense of this hypothesis came from a study showing that, during apoptosis, giantin is more stably associated with Golgi fragments than other golgins [47,48]. In addition, the Warren group, using a microsurgery approach, developed Golgi-free cytoplasts from large African green monkey cells [49], and these cells failed in de novo Golgi biogenesis. However, when cells were treated with BFA, followed by microsurgery, even a small amount of giantin in the cytoplast was sufficient to restore Golgi membranes. In addition, it was recently shown that depletion of giantin resulted in more dispersed Golgi stacks after treatment with Nocodazole, a microtubule depolymerization agent [50]. Thus, it appears that giantin is the core Golgi protein that drives Golgi biogenesis, but the underlying mechanism remained enigmatic. 

Recently, we revealed that ethanol (EtOH) treatment blocks activation of SAR1A GTPase, thus preventing COPII vesicle-mediated Golgi targeting of protein disulfide isomerase A3 (PDIA3), the chaperone that catalyzes dimerization of giantin [13,34]. In EtOH-treated hepatocytes, Golgi is consequently disorganized and Golgi targeting of mannosyl (α-1,3-)-glycoprotein beta-1,2-N-acetylglucosaminyltransferase (MGAT1), the key enzyme of N-glycosylation, is altered [51]. In addition, we previously showed that giantin determines Golgi localization for core 2 O-glycosylation enzymes (both N-acetylglucosaminyltransferase 2/M and 1/L, C2GnT-M and C2GnT-L) [34,43], and glycogen synthase kinase β (GSK3β) [52]. Moreover, we recently found that EtOH-induced Golgi disorganization in prostate cancer (PCa) cells is accompanied by mislocalization of the N-glycosylation enzyme, N-acetylglucosaminyltransferase-III (MGAT3), again implying its sensitivity to giantin [53]. Thus, different classes of Golgi residential enzymes seem giantin-dependent: recent observation in the giantin knockout model reveals another Golgi protein, N-acetylgalactosaminyltransferase 3 (GALNT3), which is down-regulated in cells lacking giantin [54]. In addition, the siRNA-mediated KD of giantin results in an abnormal rate and glycosylation of some plasma membrane (PM)-associated anterograde cargo [50]. Importantly, we also recently reported that giantin is required for post-alcohol recovery of Golgi in liver cells and PM-directed trafficking of important hepatic proteins [55]. 

Here, using different approaches, we found that BFA-induced Golgi disorganization is associated with the monomerization of giantin. In cells recovering from BFA, giantin, but not other golgins and GRASPs, is the key protein responsible for the restoration of juxtanuclear and flattened Golgi. The fusion of the nascent Golgi membranes is assisted by Rab6a GTPase. The ability to assess a Golgi 3D-structure using high-resolution microscopy enabled us to discover the sequential Golgi targeting of residential enzymes: giantin-independent proteins filled Golgi membranes during recovery, while giantin-sensitive did so only after a complete restoration of stacks. Finally, we found that giantin is involved in the formation of “long” intercisternal communications.

## 2. Material and Methods

### 2.1. Antibodies and Reagents

The primary antibodies used were: (a) rabbit polyclonal—giantin (Novus Biologicals, Centennial, CO, USA, NBP2-22321), giantin (Abcam, Cambridge, UK, ab24586 and ab93281), NMIIA (Abcam, ab75590), and GST (Abcam, ab19256); (b) rabbit monoclonal—SAR1B (Abcam, ab155278), GAPDH (Cell Signaling Technology, Danvers, MA, USA, 14C10), Man-I (Abcam, ab140613), MGAT1 (Abcam, ab180578), GM130 (Abcam, ab52649), GRASP65 (Abcam, ab174834); (c) mouse monoclonal—HSP70 (Abcam, ab2787), NMIIB (Abcam, ab684), GRASP65 (Santa Cruz Biotechnology, Dallas, TX, USA, sc365434), β-actin (Sigma-Aldrich, St. Louis, MO, USA, A2228), giantin (Abcam, ab37266), SAR1A (Abcam, ab77029), Rab6a (Santa Cruz Biotechnology, sc-81913); and (d) mouse polyclonal—MGAT1 (Abcam, ab167365), GM130 (Abcam, ab169276). The secondary antibodies (Jackson ImmunoResearch, West Grove, PA, USA) were: (a) HRP-conjugated donkey anti-rabbit (711-035-152) and donkey anti-mouse (715-035-151) for W-B; and (b) donkey anti-mouse Alexa Fluor 488 (715-545-150) and anti-rabbit Alexa Fluor 594 (711-585-152) for immunofluorescence. Brefeldin A (EMD Chemicals, Gibbstown, NJ, USA) and MG-132 (Frontier Scientific, Logan, UT, USA) were dissolved in DMSO immediately before use. Brefeldin A was added to the cultured cells at a final concentration of 36 μM, which was followed by incubation at 37 °C for the 1 h. To study Golgi recovery after Brefeldin A treatment, cells were rinsed at least 3 times with pre-warmed drug-free medium followed by incubation under regular culture conditions for various durations as described in each experiment. NEM (N-ethylmaleimide) was obtained from Thermo Scientific (Waltham, MA, USA), dissolved in water and applied to the cell pellet in at 2 mM prior to the cell lysis buffer. All other chemicals and reagents including methanol and DMSO were of MS-grade/analytical grade and purchased from Sigma.

### 2.2. Immunoprecipitation (IP), Plasmid Constructions and Transfection

For identification of proteins in the complexes pulled down by IP, confluent cells grown in a T75 flask were washed three times with 6 mL PBS each, harvested by trypsinization, and neutralized with soybean trypsin inhibitor at a 2× weight of trypsin. IP steps were performed using Pierce Co-Immunoprecipitation Kit (Thermo Scientific) according to manufacturer instructions. Mouse and rabbit non-specific IgG was used as non-specific controls. All cell lysate samples for IP experiments were normalized by appropriate proteins. To determine whether the target protein was loaded evenly, input samples were preliminarily run on a separate gel with different dilutions of control samples vs. treated, and then probed with anti-target protein Abs. The intensity of obtained bands was analyzed by ImageJ software, and samples with identical intensity were subjected to IP. MYH9 (myosin, heavy polypeptide 9, non-muscle, NMIIA), MYH10 (myosin, heavy polypeptide 10, non-muscle, NMIIB), SAR1A, SAR1B, GOLGB1 (giantin), GOLGA2 (GM130), GORASP1 (GRASP65), Rab6a, and scrambled on-targetplus smartpool siRNAs were purchased from Santa Cruz Biotechnology. All products consisted of pools of three target-specific siRNAs. Cells were transfected with 100 nM siRNAs using Lipofectamine RNAi MAX reagent (Life Technologies, Carlsbad, CA, USA). PCMV-intron myc Rab6 T27N was a gift from Terry Hebert (Addgene, Cambridge, MA, USA, plasmid # 46782) [56]. Transient transfection of cells was carried out using Lipofectamine 3000 (Life Science technologies) following manufacturer protocol. Rab6a–pCMV3–C–OFPSpark (RFP tag) was ordered from Sino Biological. GOLGB1 (giantin)–pCMV6–AC–GFP plasmids were obtained from OriGene.

The Cys3254Ser substitution in giantin protein was performed using the standard cloning and PCR-mediated site-directed mutagenesis (SDM) procedures. Given that the original plasmid GOLGB1 (giantin)–pCMV6–AC–GFP is too big for amplification (16 kb), we generated a smaller (8 kb) substrate for SDM as follows: 4 kb C-terminal fragment of the GOLGB1 from the GOLGB1 (giantin)–pCMV6–AC–GFP was subcloned into pET28b vector within EcoRV and NotI restriction sites. Then resulting plasmid pET28b-GOLGB1-C-terminus was amplified with mutagenic overlapping primers: Ser3254F (5′-GCTCATTCTGTCTTTTACGGGCCATCTAACGCGTACGCGG) and Ser3254R (5′-CCGTAAAAGACAGAATGAGCAGGACATGAATCATTAGAAAGTAGATGGCTGC). For SDM PCR we used Phusion High-Fidelity DNA Polymerase (Thermo Scientific) and cycler program: 95 °C 2′ + 15 × [95 °C 30″ + 60 °C 1′ + 72 °C (12′ + 6″)] + 72 °C 6′. After PCR completion, the PCR reaction mixture was treated with DpnI restriction enzyme (to digest methylated template) and then used to transform *E. coli* TOP10 strain. A positive clone was confirmed by restriction analysis and Sanger sequencing. Then, mutated plasmid was digested with EcoRV, NotI, and PvuI restriction enzymes. PvuI was used to cut pET28b backbone which has same (4 kb) size as subcloned C-terminus of the GOLGB1. A 4 kb EcoRV NotI fragment of the pET28b-GOLGB1-C-terminus-MUT was ligated with 12 kb EcoRV NotI fragment of the GOLGB1 (giantin)–pCMV6–AC–GFP. Positive clones were selected by restriction analysis and sequencing.

### 2.3. In Vitro Crosslinking

The protocol of crosslinking was followed according to the manufacturer’s (Thermo Scientific) instructions. Briefly, PBS-washed (three times) microsomal fraction of cells were exposed to 0.2 mM dithiobis (sulfosuccinimidylpropionate) (DTSSP) in water for 30 min at room temperature. Cross-linked protein was analyzed by SDS-PAGE under non-reducing conditions since the DTSSP cross-linker is thiol-cleavable.

### 2.4. Confocal Immunofluorescence Microscopy

The staining of cells was performed by methods described previously [29]. Slides were examined under a Zeiss 510 Meta Confocal Laser Scanning Microscope and LSM 800 Zeiss Airyscan Microscope (Carl Zeiss Microscopy, Jena, Germany) performed at the Advanced Microscopy Core Facility of the University of Nebraska Medical Center. Fluorescence was detected with fixed exposure time, using an emission filter of a 505–550 nm band pass for green, and a 575–615 nm band pass for red. Images were analyzed using ZEN 2.3 SP1 software. For some figures, image analysis was performed using Adobe Photoshop and ImageJ. Statistical analysis of colocalization was performed by ImageJ, calculating the Pearson correlation coefficient [57].

### 2.5. Three-Dimensional Structured Illumination (3D-SIM) Microscopy and Image Analysis

SIM imaging of Golgi ribbons was performed on a Zeiss ELYRA PS.1 super-resolution scope (Carl Zeiss Microscopy) using a PCO.Edge 5.5 camera equipped with a Plan-Apochromat 63 × 1.4 oil objective. Optimal grid sizes for each wavelength were chosen according to manufacturer recommendations. For 3D-SIM, stacks with a step size of 110 nm were acquired sequentially for each fluorophore, and each fluorescent channel was imaged with three pattern rotations with three translational shifts. The final SIM image was created using modules built into the Zen Black software suite accompanying the imaging setup. Analyses were undertaken on 3D-SIM datasets in 3D using IMARIS versions 7.2.2–7.6.0 (Bitplane AG, Zurich, Switzerland). The calculation of intercisternal distances was based on nearest neighbor distances to consider the Nyquist limited resolution, which in our case was around ~94 nm [58]. The 3D mask was obtained by applying a Gaussian filter to merged channels, thresholding to remove low-intensity signals, and converting the obtained stack into a binary file that mapped all voxels of interest for coefficient calculation. For colocalization studies, IMARIS “Colocalization Module” was used. To avoid subjectivity, all thresholds were automatically determined using algorithms based on the exclusion of intensity pairs that exhibit no correlation [59]. Colocalization was determined by Pearson’s coefficient, which represents a correlation of channels inside colocalized regions. After calculation, colocalization pixels were displayed as white. 3D animation was generated using IMARIS “Animation Module”.

### 2.6. AFM Imaging and Image Analysis

Giantin-GFP was isolated from DMSO and BFA-treated cells using GFP-Trap Magnetic Agarose (ChromoTek, Planegg, Germany) according to the manufacturer’s recommendations. Eluted IP samples were isolated using Millipore UFC500324 Amicon Ultra Centrifugal Filters and then dissolved in PBS for pH neutralization. Next, about 10 μL samples were treated with 2% of β-mercaptoethanol and deposited onto a piece of freshly cleaved mica. After 2 min incubation samples were rinsed briefly with several drops of deionized water and dried with a gentle flow of argon. Images were collected with MultiMode Nanoscope IV system (Bruker Instruments, Santa Barbara, CA, USA) in Tapping Mode at ambient conditions. Silicon probes RTESPA-300 (Bruker Nano Inc., Billerica, MA, USA) with a resonance frequency of ~300 kHz and a spring constant of ~40 N/m were used for imaging at scanning rate for about 2.0 Hz. Images were processed using the FemtoScanOnline software package (Advanced Technologies Center, Moscow, Russia).

### 2.7. In Situ Proximity Ligation Assay

The assay was performed using the Duolink kit (Sigma-Aldrich) according to the manufacturer’s protocol. The mouse monoclonal anti-giantin Ab, which recognizes the cytoplasmic N-terminus (3–91 aa) (Sigma, WH0002804M1) was used in combinations with: (a) rabbit polyclonal, which is raised against the central region of giantin (1781–1907 aa) (Sigma, HPA011008); and (b) rabbit polyclonal to giantin (region within N terminal 108–157 aa, Abcam ab93281). After incubation with primary Abs, cells were incubated with oligonucleotide-conjugated anti-mouse minus and anti-rabbit plus proximity ligation assay secondary probes. Subsequent PLA signal was detected by confocal microscopy and the corrected integrated fluorescence intensity was calculated using the ImageJ software.

### 2.8. Isolation of Golgi Membrane Fractions by Sucrose Gradient Centrifugation

Golgi membrane fractions were isolated using methods described previously [29]. Cells from ten-to-twelve 75 cm^2^ cell culture flasks were harvested with PBS containing 0.5× protease inhibitors (1.2 mL per flask). Then, after centrifugation for 5 min at 1000 rpm and 4 °C, the pellet was resuspended in 3 mL of homogenization buffer (0.25 M sucrose, 3 mM imidazole, 1 mM Tris-HCl; pH7.4, 1 mM EDTA). Cells were homogenized by drawing ~ 30 times through a 25-gauge needle until the ratio between unbroken cells and free nuclei became 20%:80%. The postnuclear supernatant was obtained by centrifugation at 2500 rpm and 4 °C for 3 min, and then, the supernatant was adjusted to 1.4 M sucrose by addition of ice-cold 2.3 M sucrose in 10 mM Tris-HCl (pH 7.4). Next, 1.2 mL of 2.3 M sucrose at the bottom of the tube was overlaid with 1.2 mL of the supernatant adjusted to 1.4 M sucrose followed by sequential overlay with 1.2 mL of 1.2 M and 0.5 mL of 0.8 M sucrose (10 mM Tris-HCl, pH 7.4). Gradients were centrifuged for 3 h at 38,000 rpm (4 °C) in an SW40 rotor (Beckman Coulter, Brea, CA, USA). The turbid band at the 0.8 M/1.2 M sucrose interface containing Golgi membranes was harvested in ~500 µL aliquot by syringe puncture. The fraction at a concentration of ~1.0–1.4 mg protein/mL was used for the experiments mentioned in the Section 3.

### 2.9. Zonal Sedimentation Analysis on Sucrose Gradients

Dimeric and monomeric giantin was analyzed by ultracentrifugation on layered 5–25% (*wt*/*vol*) sucrose gradients (9 mL) with a 2-mL 60% sucrose cushion prepared in 50 mM Tris-HCl (pH 7.5) and 100 mM NaCl. Isolated Golgi fractions were loaded on the top of separate sucrose gradients and centrifuged in an SW41 rotor (Beckman Coulter Inc., Fullerton, CA, USA) at 35,000 rpm for 16 h at 4 °C. Fractions of 0.25 mL were collected from the top of the tube and analyzed by immunoblotting with giantin Ab. As high molecular weight standard protein, we used an 800 kDa protein Nesprin, for which the specific band on SDS-PAGE was used for consideration of the size of giantin dimer. Densitometric analysis of the obtained band was performed using the ImageJ software.

### 2.10. Isolation of Microsomal Fraction

Isolation of microsomes was performed using Endoplasmic Reticulum Isolation Kit (Sigma) according to the manufacture’s protocol. Briefly, cells were suspended in Hypotonic Extraction Buffer and incubated for 20 min at 4 °C to allow the cells to swell. Then, after centrifugation at 600× *g* for 5 min and homogenization, the homogenate was centrifuged at 1000× *g* for 10 min (4 °C). The obtained postnuclear supernatant was centrifuged at 12,000× *g* for 15 min (4 °C), and supernatant (post mitochondrial fraction) was further subjected to isolation of rough endoplasmic reticulum (RER) using precipitation by 8 mM CaCl_2_.

### 2.11. Statistical Analysis

Measurements for the giantin KD do not follow a normal distribution as detected via the Lilliefors test at *p* ≤ 0.05 [60]; hence, hypothesis testing for different medians was performed via the non-parametric Wilcoxon rank sum test [61] for control versus giantin KD for the three parameters: cisternal length, intercisternal distance, and number of intercisternal connections. Data are expressed as mean ± SD. The rest of the analysis was performed using the two-sided t-test. A value of *p* < 0.05 was considered statistically significant.

### 2.12. Miscellaneous

Protein concentrations were determined with the Coomassie Plus Protein Assay (Pierce Chemical Co., Rockford, IL, USA) using BSA as the standard. Densitometric analysis of band intensity was performed using ImageJ.

## 3. Results

### 3.1. BFA-induced Golgi Disorganization Is Associated with Giantin Monomerization

Given that giantin de-dimerization was detected in EtOH-treated cells, as well as in PCa cells with fragmented Golgi phenotype [13,34], we hypothesized that the ER stress, induced by BFA, is also accompanied by giantin monomerization. We tested this possibility in HeLa cells treated with 36 μM BFA for 1 h, only because prolonged (up to 3 h) treatment with BFA is known to launch apoptosis, as indicated by the enhanced expression of caspase-3 (Appendix A), membrane blebbing, and chromatin condensation (Appendix A) [62,63]. As predicted, 1-h treatment with BFA induces Golgi membrane dissolution into the cytoplasm (Appendix A). Using sucrose gradient (5–20%) sedimentation analysis of Golgi fraction, we found that, in the 25% fraction from DMSO-treated cells, the giantin-specific signal appeared as double bands: with upper band corresponding to the 800 kDa (giantin dimer) and lower band around 400 kDa (giantin monomer) (Figure 1A). However, in BFA-treated cells, the portion of dimeric giantin was significantly reduced. Predictably, in 5% sucrose fraction, little if any dimeric giantin was detected in both DMSO- and BFA-treated samples. On the other hand, dimerization of giantin seems dispensable for its Golgi localization, as we were able to detect a large fraction of monomeric giantin in the Golgi remnants. Our data fit well with the previous observation that postulates the Golgi retention domain of giantin resides within its C-terminal sequence adjacent to the membrane-anchoring domain (positions 3059–3161 aa), but not within the luminal domain where disulfide bond is formed [64].

Next, we performed a series of SDS-PAGE, using the cell lysate samples prepared under different reducing conditions. Prior to lysis, samples were pretreated with 2 mM N-ethylmaleimide (NEM), which blocks free sulfhydryls. Indeed, as shown in Figure 1B, post-lysis dimerization of giantin was prevented in presence of NEM, especially in BFA-treated samples, which, however, is restored at the 60 min of BFA-WO (Figure 1C,D), again indicating that BFA induces de-dimerization of giantin. Of note, the level of giantin-dimer was positively correlated with the restoration of Golgi perinuclear positioning and was identical to untreated cells after 60 min of WO (data not shown). This raises the possibility that giantin dimerization occurs during Golgi de novo formation. Next, in samples prepared under 20 mM DTT, another strong reducing agent, the dimer-specific band was not detected at all, confirming that the dimerization of giantin is mediated by a disulfide bond [44,45] (Figure 1E).

We previously detected that in Panc-1 cells expressing C2GnT-M tagged with c-Myc, BFA treatment not only induces the redistribution of C2GnT-M to the ER, but also reduces its level via proteasome-mediated degradation [65], echoing observations of other groups [66,67]. In addition, it has been reported that upon BFA treatment the content of most Golgi matrix proteins, including giantin, may also be reduced [68]. Thus, one may assume that the loss of giantin dimer is rather a consequence of its proteasome-mediated degradation than unwinding of its dimeric structure [69]. Here, at this BFA dosage and timing, we could not detect a significant reduction in the giantin content but, to check this possibility, prior to BFA treatment, we pretreated cells with the proteasome inhibitor MG-132. Importantly, as we have shown before, MG-132 does not block the disorganization effect of BFA on Golgi [29]. Here, we observed that the effect of BFA on giantin in cells pretreated with MG-132 appears identical to cells treated with BFA only (Figure 1F). The data indicate that de-dimerization of giantin by BFA cannot be attributed to its degradation. Further, we performed experiments using a lower dosage of BFA (18 μM for 2 h or 3.6 μM for 10 h) and the results were identical: in both cases, Golgi disorganization was associated with giantin monomerization (Appendix A). However, at the 720 nM dosage, we could not detect the disorganization effect of BFA (Appendix A). 

The initial inference that BFA induces separation of resident Golgi proteins from matrix Golgi proteins [16,70,71] was revisited after findings indicating that glycosyltransferases and golgins were in the same tubular structures emanating from the Golgi and ER-localized BFA remnants [15,32,36]. However, giantin was partially detected in the ER, while some other matrix proteins, including GM130, were retained in these small tubulovesicular profiles [32]. Here, in the microsomal fraction isolated from BFA-treated cells, we were able to detect only monomeric giantin, and predictably, no visible signal of GM130 was detected (Figure 1G). Further, we could not find any pieces of evidence indicating that BFA affects di- or oligomerization of both GRASP65 or GM130 (Appendix A). Finally, a post-BFA ER/microsomal fraction of cells were incubated in 1 mM DTSSP, a cross-linker. As shown in Figure 1H, in presence of DTSSP, the giantin-specific signal appeared as double bands, again indicating the potential of giantin to form a dimer. Overall, the data suggest selectivity of BFA to different Golgi matrix proteins and imply that giantin can exist in its dimeric form only when localized to the Golgi. 

### 3.2. Examination of Giantin-Giantin Interaction by Proximity Ligation Assay

To validate this observation, we performed the quantitative in situ proximity ligation assay (PLA), which detects very close proximity between two proteins (below 40 nm) [72]; PLA is largely used to detect protein dimerization [73,74,75]. Briefly, cells were labeled with primary antibodies from different species that are specific for the two target domains. Then, secondary antibodies conjugated with the short oligonucleotides were added. These oligonucleotides were used to generate circular DNA, which was then amplified and tagged with a red fluorescence dye. Following DAPI staining of the nuclei, the cells were examined by confocal fluorescence microscopy. Thus, epitopes that are within 40 nm of each other are detected as a red fluorescent spot. Here, we employed two different antibodies to giantin: mouse monoclonal, which recognizes the cytoplasmic N-terminus (3–91 aa), and rabbit polyclonal, which is raised against the central region (1781–1907 aa). If giantin is considered as an extended Golgi protein with the cytoplasmic rod of ~250 nm [45], then we should not be able to detect PLA signal in control cells, since the target epitopes reside within the distance much larger than required 40 nm. However, to our surprise, we were able to detect PLA signal in control (DMSO-treated) cells (Figure 2A,B), suggesting that the N-terminal of giantin can bend due to flexibility in its central non-coiled coil regions (predicted by UniProtKB–Q14789). Thus, the cytoplasmic domain of giantin dimer is rather hooked than floppy rod-like (Figure 2C). We believe that the detected fluorescence reflects the PLA signal from both parallel chains of giantin dimer, including the potential crosslinking between 3–91 aa domain from one chain and 1781–1907 aa domain from another. Conversely, the PLA signal in BFA-treated cells was negligible compared to the control cells, implying that de-dimerization of giantin associates with extension of its cytoplasmic domain (Figure 2A–C). However, when we performed PLA using antibodies raised against epitopes localized in the distal N-terminus within 40 nm (same mouse anti-giantin 3–91 aa described above and rabbit polyclonal anti-giantin 108–157 aa), strong PLA signal was detected in both DMSO- and BFA-treated cells (Appendix A). Predictably, in control cells, fluorescent punctae were localized in perinuclear area, while in BFA-treated cell they were predominantly distributed at the cell periphery. Moreover, fluorescence intensity was reduced after BFA, again indicating that the part of the PLA signal in control cells can be attributed to the crosslinking interaction between the chains of giantin dimer.

### 3.3. AFM Characterization of Giantin Protein

To visualize the monomerization of giantin in BFA-treated cells, we performed the Atomic Force Microscopy (AFM) imaging. First, we attempted to isolate endogenous giantin using anti-giantin Ab coupled epoxy beads (Dynabeads M-450); however, after elution, IP fractions were contaminated with the anti-giantin IgG, making interpretation of AFM images difficult. Therefore, we decided to pull down the exogenously expressed giantin, using the GOLGB1 (giantin)–pCMV6–AC–GFP construct. Predictably, in HeLa cells, giantin-GFP localizes in the Golgi area, redistributes to the cytoplasm after BFA treatment and returns to the Golgi after BFA-WO (Appendix A). In addition, we selected this strategy because giantin-GFP was successfully isolated using GFP-Trap magnetic beads (Appendix A). These beads consist of an anti-GFP nanobody, which represents only a single variable domain (V_H_H) and can be easily separated from giantin-GFP. When we analyzed giantin-GFP by AFM, we noticed that protein forms a large complex, particularly in DMSO-treated cells, despite the fact that the lysate for these samples was prepared in the presence of NEM. This suggests that giantin-GFP is still able to aggregate into a large polymeric complex and prompted us to treat samples with a low reducing agent, 2% β-mercaptoethanol. Figure 2D shows one of the typical AFM topography images of giantin protein from BFA-treated cells obtained after deposition on the mica substrate. Proteins appear to assume globular conformations rather than extended ones, implying the changes in native conformation after GFP-Trap. This morphology can be clearly identified in the image as the bright spots. These bright blobs are quite uniform exhibiting slight variation in both the height and the diameter of the globular structures. Statistical analysis of 231 proteins resulted in <h> = 2.5 ± 0.2 nm as the height and <d> = 14.4 ± 2.9 nm as the diameter of the globular structures. Next, we used these results to estimate the volume of the proteins by using a “spherical cap” model, which has been successfully used in previous AFM studies of proteins and protein-DNA complexes [76,77,78,79]. Again, the statistical histogram of those computed protein volumes, as shown in Figure 2E has a narrow distribution, suggesting that the stoichiometry of the protein is uniform. Fitting this histogram with Gaussian function revealed two main peaks with maxima at 194 ± 5 nm^3^ and 418 ± 19 nm^3^. The value of the second peak is doubled compare to the first peak. Additionally, it comprises only ~10% of the entire population of the observed proteins. We assign these peaks to the monomeric and dimeric form of the protein, respectively.

Next, we proceed to the analysis of giantin-GFP obtained from DMSO-treated cells. We noticed a greater size variation of globular structures, as shown in the representative image (Figure 2F). The appearance of small globules resembles the ones observed in BFA-treated sample, which does not seem surprising given the presence of β-mercaptoethanol. However, along with small globules, we observed the larger ones, indicating that in control cells giantin exists in a mixture of monomeric and multimeric complexes. Statistical analysis of volumes for 457 proteins produced a histogram shown in Figure 2G. Fitting this histogram with Gaussian function produced several peaks. The first peak has the same maximum as the monomeric BFA-specific giantin at V = 198 ± 9 nm^3^. The second peak has a maximum at 445 ± 21 nm^3^, which again is almost double the maximum of the monomeric form. This peak comprises 27% of the total population of proteins analyzed. These results confirm that giantin from DMSO-treated cells forms more dimer than that from cells treated with BFA. Surprisingly, there are also larger multimers observed for both BFA- and DMSO-specific giantin. However, while there are only a few multimers present in images of the BFA-treated sample, the relative population of giantin-multimers is much larger in DMSO-treated samples. These results imply that giantin can potentially form oligomeric complexes. Indeed, in the SDS-PAGE performed under non-reducing condition, we observed in DMSO-treated samples giantin-specific band, whose size larger than expected 800 kDa of the dimer (Figure 2H, asterisk). Notably, in BFA-treated samples, this band was not visible, echoing the results presented in Figure 2D,E.

### 3.4. Giantin Dimerization Is Critical for Golgi Biogenesis

It has been suggested that giantin forms a dimer via disulfide-linked lumenal domain at Ser3254 [44,45]. To check the critical role of this link in homodimerization of giantin and reformation of Golgi, we constructed giantin-GFP mutant protein by substitution of the Cys3254 residue with Ser, using the GOLGB1 (giantin)–pCMV6–AC–GFP plasmid mentioned above. Analogously to the wild-type (WT) gaintin-GFP, giantin(-C3254S)-GFP was detected in the Golgi (Figure 3A,B) of HeLa cells. This was anticipated given that this mutation does not interfere with the Golgi retention signal [64]. While cells overexpressing giantin-WT-GFP were able to reform compact and perinuclear Golgi after BFA-WO, cells transfected with mutant giantin(-C3254S)-GFP failed to do so, according to the immunostaining with *cis*-Golgi GM130 and *medial*-Golgi MGAT1 (Figure 3C–E). Next, we performed GFP IP form the lysates of cells overexpressing gaintin-GFP or giantin(-C3254S)-GFP and found that contrary to the WT form, giantin lacking Cys at 3254 position does not form a dimer (Figure 3F). Altogether these data indicate not only the critical role of this disulfide link in the dimerization of giantin but also the leading contribution of the latter to Golgi biogenesis.

### 3.5. Giantin Dictates Sequential Targeting of Golgi Residential Enzymes in BFA Washout Cells

It has been suggested that, during post-BFA Golgi biogenesis, Golgi matrix proteins form a dynamic framework for subsequent delivery of glycosylation enzymes. Accumulation of the Golgi resident proteins in the nascent Golgi stacks was suggested to be sequential, starting from the *trans*-proteins and followed by their *cis-medial* counterparts [80,81]. We recently found that the *cis*-Golgi protein, α-1,2-mannosidase (Man-I) may use GM130-GRASP65 as the Golgi docking site, whereas the next to Man-I enzyme in the N-glycosylation pathway, *medial*-Golgi glycosyltransferase MGAT1, appeared giantin-responsive [51]. Given that GM130 and giantin reside predominantly in the *cis*- and *medial*-Golgi, respectively [43,82,83], we analyzed in BFA-treated HeLa cells the dynamics of colocalization of Man-I and MGAT1 with GM130 and giantin, accordingly. In untreated cells, the vast majority of Man-I and MGAT1 immunofluorescence (IF) was detected in the Golgi (Figure 4A,B). BFA-induced Golgi dissolution is accompanied by significant segregation of Man-I and MGAT1 from GM130 and giantin, respectively (Figure 4A,B); however, we should admit that the minor colocalization of resident enzymes with Golgi matrix proteins was always detectable after BFA. In the meantime, in BFA-WO cells, Man-I, but not MGAT1, was detected in recovered Golgi membranes, but the restoration of MGAT1′s intra-Golgi signal occurred only after complete recovery of the Golgi (Figure 4A,B,E). This MGAT1-related phenomenon was validated using GM130 as an alternative marker of Golgi membranes (Appendix A). Next, to better evaluate the distribution of golgins and resident proteins in reforming Golgi membranes, we decided to employ structured illumination superresolution microscopy (SIM) to create 3D reconstructed images with a lateral resolution (~110 nm) approximately twice that of diffraction-limited instruments. The calculated Pearson coefficient of colocalization confirmed that, contrary to Man-I, MGAT1 was segregated from the emerging Golgi membranes (Figure 4C,D,F and Appendix A).

These data raise the possibility that giantin-sensitive enzymes may populate Golgi at the latest stages of Golgi biogenesis. To find the definitive argument to support this concept, we performed the series of PLA. Using a combination of Abs ((a) mouse anti-MGAT1 and rabbit anti-giantin; and (b) mouse anti-GM130 and rabbit anti-Man-I), we detected a strong PLA signal in control cells, which was significantly reduced after BFA treatment (Figure 5A,B), echoing the results presented in Figure 4. However, after 30 min of BFA recovery, PLA fluorescence was recovered only in cells exposed to GM130 + Man-I Abs (Figure 5A,B), again confirming the physical closeness of these proteins (but not MGAT1 and giantin) in emerging Golgi membranes.

While we have shown previously that in cells depleted from giantin, MGAT1 was mislocalized to the ER [51], we have not yet clarified whether MGAT1 directly interacts with giantin. Here, we performed several experiments to explore the potential mechanism of MGAT’s docking at the giantin-specific Golgi site. First, we detected that giantin Ab was able to pull-down MGAT1 from the Golgi fraction of control HeLa cells (Figure 5C). In the meantime, an only marginal fraction of MGAT1 was pulled down in cells recovered for 30 min after BFA-WO; however, after 60 min of recovery, when compact Golgi appears in most of the cells (Figure 4E), the fraction of giantin-associated MGAT1 was significantly higher (Figure 5C). This finding implies that the short cytoplasmic tail of MGAT1 directly interacts with giantin. Previously, to evaluate the interaction of Golgi residential proteins with non-muscle Myosin IIA (NMIIA), we employed the N-terminal biotinylated peptides representing the cytoplasmic tail of different glycosyltransferases [29,65]. Peptides were linked to magnetic Dynabeads followed by incubation with lysate. Here, we used the same tool to examine whether endogenous giantin interacts with a biotin-MLKKQS peptide (GenicBio BioTech, China) that matches N-terminus of MGAT1. The control peptide biotin-GHGTGSTGSGSMLRTLLRRRL (LifeTein LLC, South Plainfield, NJ, USA) incubated with lysate and Dynabeads, as well as the lysate incubated with Dynabeads only served as a control. As shown in Figure 5D, Dynabeads carrying MGAT1 peptide were able to pull-down giantin from the lysate of HeLa cells, however, giantin was not detected in the pull-down fraction from the lysate exposed to the Dynabeads or in combination with control peptide.

Since we observed the positive PLA signal using mouse polyclonal anti-MGAT1 (immunogen - full-length protein, corresponding to 1–445 aa of human MGAT1) and rabbit polyclonal anti-giantin 108–157 aa (Figure 5A), it is logical to hypothesize that the MGAT1 binding domain of giantin lies within its N-terminal non-coiled-coil area (UniProtKB-Q14789). To examine this plausibility, we employed the recombinant human N-terminal GST-giantin fusion peptide (3–92 aa, Abcam). This peptide is successfully immobilized by anti-GST Ab coupled epoxy beads (Dynabeads M-450), according to the GST Western blot (W-B) (data not shown). Then, the beads containing giantin-GST was incubated (overnight, at 4 °C) with the cell lysate of non-treated HeLa cells. The anti-GST Ab coupled epoxy beads incubated with lysate only, as well as the IP by the non-specific IgG Ab coupled epoxy beads were served as a control. W-B analysis of the lysate and pull-down fractions revealed that substantial amounts of endogenous MGAT1 were pulled down by giantin-containing beads, but not by beads alone or control IgG-coupled beads (Figure 5E). Altogether, data in Figure 5A–E indicate that the cytoplasmic tail of MGAT1 is able to interact directly with the N-terminal of giantin.

We found another proof of differential Golgi targeting of O-glycosylation enzymes in PCa cells. As we described previously, Golgi transportation of C2GnT-L requires giantin and intact Golgi morphology, whereas Golgi-directed trafficking of the Core 1 synthase (C1GalT1, *cis*-Golgi) and β-galactoside α-2,3-sialyltransferase-1 (ST3Gal1, *medial-trans*-Golgi) is GM130-GRASP65-dependent and does not require a compact Golgi structure [34]. In non-treated, low passage LNCaP (c-28) cells, all three enzymes were localized to the Golgi, and, predictably, BFA treatment resulted in their relocation to the ER (Appendix A). In BFA-WO cells, both C1GalT1 and ST3Gal1 were returned to the membranes as soon as Golgi reformed back. However, the pool of C2GnT-L on the recovered Golgi was very faint (Appendix A), resembling the results of MGAT1 in HeLa cells (Figure 4). Thus, it seems that the delayed Golgi targeting of MGAT1 and C2GnT-L is associated with incomplete giantin dimerization and Golgi reconstruction. These data suggest that the genesis of Golgi architecture and enrichment of resident proteins do not necessarily coexist simultaneously. Rather, we propose a simple model illustrating that giantin-sensitive N- and O-glycosylation enzymes are packaged in their appropriate site only within a compact and perinuclear Golgi (Figure 5F).

### 3.6. Giantin, but not Other Golgi Proteins, Is the Main Driver of Golgi Biogenesis

These findings led us to the question of whether giantin is the critical protein in the formation of juxtanuclear Golgi. To answer this, we performed in HeLa cells two series of the individual siRNAs (containing the pools of three to five target-specific siRNAs) depletion of giantin, GM130, and GRASP65, followed by treatment with BFA and its WO. We found that KD of GM130 or GRASP65 does not alter Golgi reassembly; however, in cells lacking giantin, Golgi remains disorganized, even though multiple Golgi fragments intend to concentrate around the nucleus (Figure 6A,B,D). Quantitatively, depletion of giantin results in almost complete loss of ability to restore compact and perinuclear Golgi, but, in GM130- or GRASP65-depleted cells, the number of cells which failed to recover Golgi was only marginal and identical to control (Figure 6C). To validate the universality of this screen, we repeated the same experiment in Panc-1 (Appendix A), A549 (Appendix A), and LNCaP (c-28) (Appendix A) cell lines derived from different organs and confirmed that post-BFA restoration of compact Golgi is governed by giantin, but not by GM130-GRASP65 complex.

It is known that depletion of different Golgi proteins, including GRASP55 [30], GCC88 [84], Golgin-97 [85], Golgin-245 (or p230) [86], GCC185 [87], and TMF [88], may affect the Golgi ribbon and induce central Golgi fragmentation, but it has no significant effect on the perinuclear position of Golgi. To check whether any of these proteins are involved in the fusion of Golgi membranes, we performed individual siRNA-mediated depletion of their genes and monitored reconstruction of Golgi in cells recovered after 30 and 60 min of BFA-WO. While KD of GRASP55 induces moderate unstacking of Golgi membranes, it does not prevent Golgi recovery after BFA-WO (Appendix A). In cells lacking either GCC88 (Appendix A) or Golgin-97 (Appendix A), Golgi undergoes marginal enlargement, but it does not block post-BFA reformation of perinuclear Golgi. Similarly, despite the obvious but minimal changes in Golgi morphology after KD of Golgin-245 (Appendix A), GCC185 (Appendix A) or TMF (Appendix A), cells lacking any of these golgins do not lose the ability to reconnect Golgi membranes in juxtanuclear space. Importantly, none of these proteins is required for BFA-induced Golgi collapse because the effect of this drug in golgin- or GRASP55-depleted cells was identical to cells treated with control siRNAs (Appendix A).

Next, we proceeded to check the role of Rab GTPases in Golgi biogenesis, since these enzymes are required for the SNARE-mediated fusion of membranes [89]. Out of 70 human Rab GTPases, 20 proteins show Golgi localization [90] but, to the best of our knowledge, only Rab1a, Rab6a, Rab18, and Rab41 may interfere with Golgi morphology [34,91,92]. While Rab1a and Rab18 demonstrate predominantly perinuclear localization (Appendix A, accordingly), the distribution of Rab41 was detected in both cytoplasm and Golgi (Appendix A). The depletion of either Rab1, Rab18, or Rab41 has no effect on the perinuclear situation of Golgi, but, in Rab1a KD cells, we were able to detect mild enlargement of Golgi membranes. However, none of these GTPases is required for restoration of Golgi after BFA-WO (Appendix A). Based on these results, we decided to focus on the possible role of Rab6a in interconnection of Golgi membranes.

### 3.7. The Mutuality of Rab6a and Giantin

Others and we have shown that retrograde Golgi-to-ER transportation of glycosyltransferases is mediated by the interaction of their cytoplasmic tail with NMIIA, and it is coordinated by the function of Rab6a [8,29,65,93,94,95]. We recently observed that giantin and NMIIA may compete for Rab6a. In cells treated with EtOH, giantin de-dimerization was accompanied by the loss of its link to Rab6a. In the meantime, we observed the enhanced complex between Rab6a and NMIIA, which creates a force for Golgi disassembly [95]. Importantly, Rab6a was required for post-alcohol recovery of Golgi in hepatocytes [55]. We also reported that in PC-3 and DU145 cells, Golgi morphology appears disorganized, however, treatment with NMIIA inhibitor Blebbistatin or transfection with NMIIA siRNAs converts fragmented Golgi to the compact structure [34]; this Golgi “metamorphosis” was also conducted by Rab6a. Finally, it has been shown that Rab6A and giantin directly interact via their N-terminus [96]; however, the link between giantin and Rab6a in Golgi biogenesis has never been explored. Here, we examined the interaction of Rab6a and giantin, using again the model of post-BFA “Golgi renaissance”. First, in LNCaP cells after 30 min of BFA-WO, we detected less NMIIA but more giantin associated with Rab6a than in cells right after 60 min treatment with BFA (Figure 7A). Second, we confirmed this observation by detecting colocalization of Rab6a with giantin but not NMIIA in the partially reconstructed Golgi of LNCaP cells after BFA-WO (Figure 7B,C). Third, when we performed life cell imaging of BFA-WO HeLa cells that co-expressed giantin-GFP and Rab6a-RFP, the multiple giantin-Rab6a colocalizing punctae was seen in restored perinuclear structures (Appendix A).

To validate these screens, we implemented the proof-of-principle experiment, using cell-free reconstitution of Golgi membranes [97]. First, we isolated Golgi membranes from non-treated and BFA-treated HeLa cells according to the protocol established previously [29]. As shown in Figure 7D, the remnants of the Golgi membranes from BFA-treated samples contain giantin, but not Rab6a. Thus, we used these samples to study the effect of exogenous Rab6a on the fusion of the nascent Golgi membranes. Briefly, the 500 µL of Golgi membranes isolated from BFA-treated HeLa cells were exposed to 5 µg of active Rab6a protein (Abcam) in the presence of the cytosolic fraction and the Reaction Mix, which includes 20 μL of 0.5 M EDTA, 10 μL of 100× GTPγS (Cell Biolabs), and 2 μL of ATP (Energy) Regeneration Solution (Enzo). The suspension was incubated in 37 °C water bath for 15 min. Then, the reaction was stopped by 65 μL of 1 M MgCl_2_. Another sample of Golgi membranes was prepared analogously, except the addition of active Rab6a protein. Rab proteins require prenylation for insertion into membranes. Herein, we detected that, under such conditions, Rab6a protein was prenylated, since it was detected in the detergent fraction after Triton X-114 phase partitioning (data not shown) [98]. Next, samples were incubated with anti-giantin Ab followed by Alexa Fluor 488 secondary Ab (1 h at RT for each step). Then, samples were gently transferred to the slides, air-dried under the hood in dark, and covered by the glass slips using ProLong antifade mountant (Thermo Scientific). Finally, we reconstructed 3D volume-rendered surfaces from the SIM imaging of Golgi membranes and evaluated the luminal length of these membranes by the ImageJ software. In the samples lacking Rab6a, the largest segregated Golgi structures did not exceed 2 µm; however, the length of Golgi membranes that have been fused in presence of Rab6a was significantly increased and very close to the values we saw in the Golgi fraction from non-treated cells (Figure 7E,F). These data strongly indicate that the maturation of Golgi membranes and their fusion are mediated by the function of Rab6a. Indeed, in another series of 3D SIM imaging of BFA-WO HeLa cells, endogenous Rab6a and giantin were found in the close vicinity (Figure 7G, top panel). Remarkably, in many reconstructed Golgi surfaces, we detected Rab6a punctae between the two emerging giantin-stained membranes (Figure 7G, bottom panel). To further check this hypothesis, we performed the PLA experiment using mouse anti-giantin and rabbit anti-Rab6a Abs. In control cells, we detected only a moderate PLA signal. It does not seem surprising, because contrary to *medial*-Golgi giantin [43,82,83], Rab6a is preferentially distributed in the *trans*-Golgi, where is involved in different intra-Golgi events [98,99,100,101]. No physical closeness of Raba and giantin was detected in BFA-treated cells, again confirming the segregation of Rab6a from the BFA-induced Golgi remnants (Figure 7D). However, PLA punctae were identified after as early as 30 min of BFA-WO (Figure 7H,I), indicating that the strong interaction between Rab6a and giantin is detectable only in the fused Golgi membranes.

These data prompted us to check whether a mutual link exists between Rab6a and giantin, i.e., whether localization of Rab6a to the Golgi is impaired in giantin-depleted cells recovering after BFA treatment, and, conversely, whether fusion of Golgi membranes is impaired in cells lacking Rab6a. In cells treated with scramble siRNAs, after 60 min of BFA-WO, the IF signal of Rab6a was predominantly compact and perinuclear. A similar distribution of Rab6a was observed in GM130- or GRASP65-depleted cells recovered from BFA; however, in cells transfected with giantin siRNAs, after 60 min of BFA-WO, Rab6a was equally distributed throughout the cell (Figure 8A). Quantification of Rab6a IF associated with membranous structures indicated that giantin KD, but not GM130 or GRASP65 KD, blocks the recollection of Rab6a into the Golgi membranes (Figure 8B). Next, in cells treated with scramble siRNAs, after 60 min of BFA-WO, the distribution of giantin IF signal was closed to the non-treated cells (Figure 8C,D,F). At the same time, cells transfected with Rab6a siRNAs failed to recover Golgi, according to the staining with either giantin (Figure 8D–F) or GM130 (Appendix A). The similar results were obtained in cells transfected with dominant-negative (GDP-bound) Rab6a(T27N), confirming that Golgi cisternal maturation requires GTPase activity of Rab6a (Figure 8D,F and Appendix A). In sum, these data clearly indicate that Rab6a is required for the stabilization of dimeric structure of giantin and Golgi reformation.

### 3.8. Is GRASP65 the Protein that May Compensate for the Lack of Giantin?

The alternative to the giantin Golgi docking site is represented by GM130, a segmented coiled-coil dimer of which the C-terminal region binds to Golgi membranes preferentially through interaction with GRASP65 [43,102,103,104]. However, our previous results indicate that in the absence of GRASP65, GM130 may form a complex with giantin [34,43]. Given that, in cells lacking both giantin and GRASP65, the intra-Golgi signal of GM130 is compromised [43], it is logical to assume the existence of a reserved Golgi tethering mechanism for GM130, one that in absence of GRASP65 can be realized through a direct link between giantin and GM130. Our results thus far suggest that giantin is essential for the biogenesis of Golgi in terms of its compact structure and perinuclear localization. However, in giantin-depleted cells, GM130- and GRASP65-specific immunostainings do not seem significantly different from the cells transfected with control siRNAs, at least at the level of conventional multi-fluorescence microscopy [13,54,105]. This suggests that cells that are not under the treatment with Golgi disturbing agents but are experiencing a deficiency in giantin may launch a compensatory mechanism to maintain Golgi positioning and function. Among Golgi proteins, GRASP65 and GRASP55 are the only two that form stable homodimers to further oligomerize in trans-form [106,107,108]. It has been proposed that this structural feature of GRASPs is essential to hold adjacent Golgi membranes in stacks: GRASP65 for the *cis*-Golgi, and GRASP55 for the *medial-trans*-cisternae [35,109]. Remarkably, the siRNA-mediated KD of GRASP65 increases the level of giantin [43]. However, it is important to note here that the response of GRASP55 to ER stress differs from GRASP65 [110,111,112], and thus far no virtual link between giantin and GRASP55 is reported, raising the possibility that at the level of *cis-medial*-Golgi, functions of giantin and GM130-GRASP65 (but not giantin and GRASP55) may overlap, allowing mutual compensation. In HeLa cells, giantin depletion increases the content of GRASP65; however, the densitometric ratio monomer/dimer was identical to control cells. In the meantime, the level of GRASP65 tetramer was significantly enhanced (Figure 9A,B). We reasoned that giantin, and GRASP65 in absence of giantin, serve as the scaffold for Golgi intercisternal connections that seem necessary to maintain not only Golgi stacking but also rapid trafficking and processing [113,114,115]. Indeed, when we performed co-depletion of both giantin and GRASP65, Golgi perinuclear positioning and organization were substantially impaired (Figure 9C–E).

Thus, cells lacking detectable giantin appear to possess Golgi stacks and perinuclear location due to GRASP65 oligomerization. However, several cardinal differences came to our attention after thorough ultrastructural analysis of Golgi performed by a series of 3D SIM imaging. First, the average total length of contiguous Golgi cisternae decreased to 1.509 ± 0.609 µm in giantin-KD cells, from 15.12 ± 4.597 µm in control cells. The same phenomenon was detected in prostate cell lines: RWPE-1 and LNCaP (data not shown). Second, a distance of intercisternal connections was significantly reduced: the “long” communications in control cells (1.509 ± 0.6090 µm) after giantin depletion were replaced by the “short bridges” (280 ± 204 nm) (Figure 9F,G). Finally, the number of cisterna-to-cisterna communications per Golgi stack was also lower in cells lacking giantin (4.6 ± 1.578) compared to control cells (8.7 ± 1.636). Kernel density plots calculated for each variable revealed significant differences in the location and shape of the distributions by group (Appendix A). Next, the visualization of described differences was combined in an ellipsoid at the XYZ plot (Figure 9H). These results suggest that giantin appears to be critical for the maintenance of Golgi cisternae continuity and their connections.

## 4. Discussion

The ability of the Golgi apparatus to recover after severe attacks is unique and could play a significant role in cellular homeostasis. Here, we describe the role of the largest golgin giantin in the maintenance of Golgi stability and the dynamics of its membranes. We show that BFA-induced Golgi disorganization is associated with the loss of giantin dimeric structure. At first glance, it seems hard to imagine how a long dimer could unwind quickly. However, we are not the first to describe such kind of phenomenon. For example, several coiled-coil proteins, including golgin GCC185, may unwind in several domains [116,117]. In addition, some dimeric myosin motor proteins may exist in their monomeric form [118]. In addition, the induction of ER stress by the Arf1 mutant results in the relocation of GRASP55 dimer to the ER and its subsequent monomerization [111]. Notably, in different cells and organs, EtOH-induced Golgi disorganization was associated with giantin de-dimerization [13,52]. In our preliminary screening, treatment of prostate cells with the ER stress inducer, Thapsigargin, results in Golgi fragmentation, which again was accompanied by the loss of giantin dimer (manuscript in preparation). Highly aggressive PCa cells, PC-3 and DU145, which are experiencing ER stress under normal conditions, demonstrate fragmented Golgi phenotype and lack of giantin dimer [34]. Thus, it is attempting to speculate that link between giantin monomerization and Golgi disorganization can be attributed to not only the BFA-induced cellular perturbations but also to the other clinically relevant cases of ER stress. This assumption still requires rigorous investigations.

We have shown that giantin C-terminal is critical not only for its Golgi retention but also for its dimerization and Golgi reassembly. We would not exclude the potential contribution of the coiled-coil domain of giantin to the fusion of Golgi membranes. In our preliminary screening, we observed that giantin (C3254S) construct with additional potentially critical point mutation in the coiled-coil domain (Glu1978Lys) still able to reside in the perinuclear space and recover Golgi after BFA (data not presented). However, the precise role of coiled-coil domains in the stabilization of giantin dimer still remains uncovered. Our results are echoing previous observation of Warren’s group [104]; however, they do not fit completely with the publication of Misumi et al. [64], who showed that truncated giantin mutant lacking C-terminal transmembrane domain (2619–3162aa) is not only able to reach the Golgi in HeLa cells but also does not block post-BFA Golgi reassembly. Since this giantin peptide lacks a disulfide-bonded lumenal domain, it seems conflicting with the data presented in our story. However, these authors used low dosage (7 µM) and short time (30 min) treatment of BFA. Using different combinations of BFA dosage and timing, we noticed that treatment of HeLa cells with 36 µM BFA for 1 h was necessary to induce complete redistribution of giantin IF signal to the cytoplasm, when cells already exhibit no perinuclear Golgi elements, but does not undergo apoptosis yet. Thus, we prefer the model of Golgi collapse, as described by Perez group, when giantin is “transported back to the ER in the presence of BFA as the Golgi ‘blinks out’” [119]. Quite contrarily, in their figures, Misumi et al. presented BFA-treated cells that still demonstrate predominant disposition of giantin punctae in the perinuclear area.

Our results clearly indicate that Golgi biogenesis requires giantin, and this finding harmonizes well with recently published data by Stevenson et al. that the recovery of Golgi after BFA was abolished in the giantin KO model [54]. While we observed the reduction of cisternal length in cells lacking giantin, these authors could not detect significant changes in this Golgi parameter. Moreover, another publication by Satoh et al. [120] observed that loss of giantin elongates Golgi cisternae, contradicting their own previous observation [50]. There are several explanations for this discrepancy. First, given that individual Golgi cisternae are complex structures that are extensively interconnected, we used 3D reconstructed images of Golgi, while authors of these two publications analyzed the mean length of cisternae from separate Z-stacks obtained by electron microscopy (EM). Second, the conclusion of these groups was based on the calculations for all Golgi cisternae, including *trans*-Golgi and elongated membranes of the *trans*-Golgi network. Instead, we are focused on the *cis-medial*-Golgi, where GRASP65 and giantin mostly reside. Finally, the difference between our data and results from Satoh et al. can be partially ascribed to the variety of sizes found in HeLa cells. Indeed, using the EM technique, two different groups presented contradicting results regarding the effect of double knockdown of GRASP65/55 on unstacking of Golgi in HeLa cells [109,121].

Here, we found that the reformation of Golgi membranes also depends on the activity Rab6a GTPases. The appearance of Rab6a in the Golgi coincides with giantin dimerization and Golgi reconstitution. At this point, we may assume the existence of at least two events that require the virtual involvement of giantin. First, during clustering to the perinuclear region, giantin monomers from the rims of opposite cisternae form the dimer via disulfide bond in the luminal domain. Then, giantin’s and Rab6a’s N-terminus tether to each other, thus forming dual giantin-Rab6a dimeric complexes (Figure 10); given the ability of Rab6a to dimerize [95,122], this scenario seems realistic. This, in turn, stabilizes giantin dimer via the formation of a coiled-coil structure. We believe that this is the critical step in the fusion of Golgi membranes because cells lacking Rab6a or its GTPase activity are unable to recollect Golgi membranes in the perinuclear space. Second, giantin seems indispensable for the formation of a communication between Golgi stacks. Our results also indicate that, in addition to the dimeric form, giantin may exist as an oligomer (Figure 2). The nature of this phenomenon remains unknown and could be subject to future studies that could also address the question of whether giantin oligomeric complexes are essential for the formation of large cisterna-cisterna communications. While the function of these internal Golgi “bridges” remains elusive, it was suggested that they could play a role in intra-Golgi trafficking and maintenance of the compact shape of the Golgi [113,114,115]. The alternative “short” intercisternal communications provided by GRASP65 oligomers seems efficient for Golgi compaction and positioning, but not sufficient for proper processing, because giantin depletion cardinally changes rate and quality of glycosylation, at least in HeLa cells [50]. It is important to note here that, in normal cells, giantin depletion itself does not lead to the loss of Golgi centralization, because the gradual decline in giantin expression can be compensated by the oligomerization of GRASP65. However, giantin appears to be critical for the recovered cells that attempt to restore Golgi architecture after different stresses.

It is known that cells pretreated with NMIIA siRNA or its inhibitor Blebbistatin demonstrate a significant delay in BFA-induced Golgi disorganization [65,123]. We have also shown that NMIIA is tethered to Golgi membranes under different drug and stress conditions, such as treatment with EtOH, heat shock, or inhibition of heat shock proteins (HSPs), and depletion of the beta-COP subunit of COPI [8,29,95]. Thus, it seems that NMIIA is a master regulator of ER stress mediated Golgi disorganization. However, we recently found that post-alcohol recovery of compact Golgi in hepatocytes is blocked in cells depleted from non-muscle Myosin IIB (NMIIB), the isoform of NMIIA [55]. Our preliminary data (data not shown) indicate that NMIIB is also required for post-BFA Golgi recovery, implying that intercisternal fusion requires a force that can be created by the action of NMIIB. Although there exists no direct evidence as to how NMIIB is tethered to the Golgi, our data make this a likely scenario.

The relationship between the matrix and resident Golgi proteins is increasingly viewed as a mutual alliance, where the function of some may influence the behavior of others [30,31,43,94,124,125]. On the one hand, golgins and GRASPs are able to form a Golgi matrix framework regardless of the presence of Golgi enzymes [71]. On the other hand, Golgi appears as a disassembled structure when some glycosyltransferases fail to localize to the Golgi [126,127,128]. For example, a mutation in the membrane-spanning domain of an N-acetylglucosaminyltransferase I caused a dramatic effect on Golgi morphology [129]. In CHO cells lacking N-acetylglucosaminyltransferase V (MGAT5), the Golgi volume density was significantly reduced [130]; however, this was not ascribed to the lack of enzymatic function, given that in the absence (or inhibition) of its precursors, MGAT1 and Mann-II, the Golgi seems unaffected. However, we do not rule out that this possibility may be limited only to the glycosyltransferases that form complexes with cytoskeletal proteins [131,132,133]. It is important to note, however, that the cytoplasmic tail of Golgi enzymes plays an essential role not only in their Golgi retention but that it also mediates tethering of NMIIA to the Golgi membranes, thus providing the force for alcohol- or BFA-induced Golgi disorganization [29,65,95,133,134]. Here, we provide the evidence which indicates that the cytoplasmic domain of MGAT1 is also required for the Golgi targeting. Since the N-terminal of giantin is projecting into the cytoplasm, it is clear that the direct interaction between MGAT1 and giantin occurs outside of the Golgi lumen. However, we still do not know whether giantin, in addition to the docking function, can also serve as a Golgi retention partner for MGAT1. This possibility requires further investigation.

Our results confirm that the response of giantin to rapid ER stress is distinct from that of GM130 and GRASP65: while some golgins (including Golgin-45 and giantin) and GRASP55 can be partially detected in the ER, both GM130 and GRASP65 are still retained in the Golgi remnants [110,111,112]. Moreover, we observed that BFA treatment does not impair the oligomerization of these proteins and provided evidence that the function of the GM130-GRASP65 complex is not critical for the Golgi recovery. This may explain why GM130-GRASP65-dependent enzymes are able to quickly refill emerging Golgi membranes. Our model suggests that giantin dimerization is a necessity for successful Golgi biogenesis. Since the formation of giantin-dimer requires functional COPII [51] and, broadly speaking, the dynamic of Golgi membranes is linked to the integrity of ER exit site [36,135], we believe that any substantial disturbance of ER function would inevitably result in Golgi dysfunction and its subsequent disorganization.

We show here that in cells lacking giantin, GRASP65 oligomerization may take a leading role in Golgi remodeling and maintenance of its positioning. However, GRASP65 consistently fails to form oligomers in different PCa cells with fragmented Golgi phenotype (our manuscript, in press). We would not exclude the possibility that cancer-specific Golgi fragmentation [136,137,138,139,140] is influenced by the dysfunction of ER, because multiple studies observed the virtual link between ER stress and cancer [141]. Therefore, prolonged but sub-lethal ER stress may lead to not only impairment of giantin dimerization, but also to alteration of posttranslational modification of GRASPs and other golgins [142]. In this case, the consequences of Golgi disintegration will appear more severe than one induced by the deficiency in giantin dimerization after treatment with BFA. Lastly, most golgins and GRASPs have multiple potential N- or O-glycosylation sites, and in the case of giantin, this was preliminary confirmed [34], suggesting their posttranslational modification may require the assistance of Golgi enzymes. Thus, it is reasonable to conclude that full maturation of Golgi matrix proteins occurs after the appearance of all resident enzymes at the Golgi membranes and complete Golgi recovery. For example, in parasitic protozoan *Giardia lamblia*, Golgi was not identified in nonencysting cells; nevertheless, the induction of Golgi enzyme activities coincides with the stacking of Golgi during differentiation to cysts [143]. Therefore, our model requires that some of the proteins (which Golgi targeting likely requires giantin) prefer to hold the occupation of membranes until complete recovery of Golgi (Figure 5F). The appearance of MGAT1 in the reformed Golgi membranes coincides with giantin dimerization, which prompts us to speculate that the binding site of giantin for MGAT1 becomes available only after giantin dimerization. This assumption requires further experimental support. In some aspects, this conception does not fit with the idea that enzymes refill membranes during Golgi biogenesis according to their intra-Golgi (from *trans* to *cis*) location [80,81]. However, given that early forming structures in Golgi assembly are inactive in cargo transport [81], the holdup of giantin-sensitive enzymes seem reasonable, because, in this scenario, the accomplishment of Golgi recovery would coincide with the commencement of protein glycosylation.

If one assumes that, in mammalian cells, compact and perinuclear Golgi is essential for complete glycosylation, does it follow that any disturbance in Golgi morphology will result in the abnormal glycan processing of cargo? The answer lies in the comparison of BFA with other Golgi disruptive agents, such as Cytochalasin D or Nocodazole. Indeed, destabilization of actin by Cytochalasin D [144] or alteration of microtubules after Nocodazole [145] results in extensive Golgi fragmentation [29]. Nevertheless, in these cells, the giantin-dependent enzyme, for instance, C2GnT-M, still localizes in the Golgi [29], suggesting that these chemicals have no significant effect on the structure or function of giantin. This could explain the abnormal glycosylation in BFA-treated cells [146,147] that is not found in cells incubated with either Cytochalasin D or Nocodazole [30,148]. Therefore, while Golgi is anchored in juxtanuclear space inter alia by cooperation with cytoskeleton proteins, successful glycosylation is determined by the intra-Golgi location of residential enzymes rather than by the positioning of Golgi [149,150]. The important question of how these separated “mini-Golgi”, induced by either Nocodazole or Cytochalasin D, communicate to maintain glycosylation of cargo remains to be answered. At first glance, clues may come from the vesicular Golgi model rather than a cisternal maturation conception; however, we are still far from a complete understanding of the nature of the many COPI- and COPII-independent vesicular and tubular complexes that have been detected in the Golgi [43,65,151,152,153]. One of the promising arsenals for these studies is high-resolution microscopy, without which accomplishment of the current story would have been impossible.

## Figures and Tables

**Figure 1 cells-08-01631-f001:**
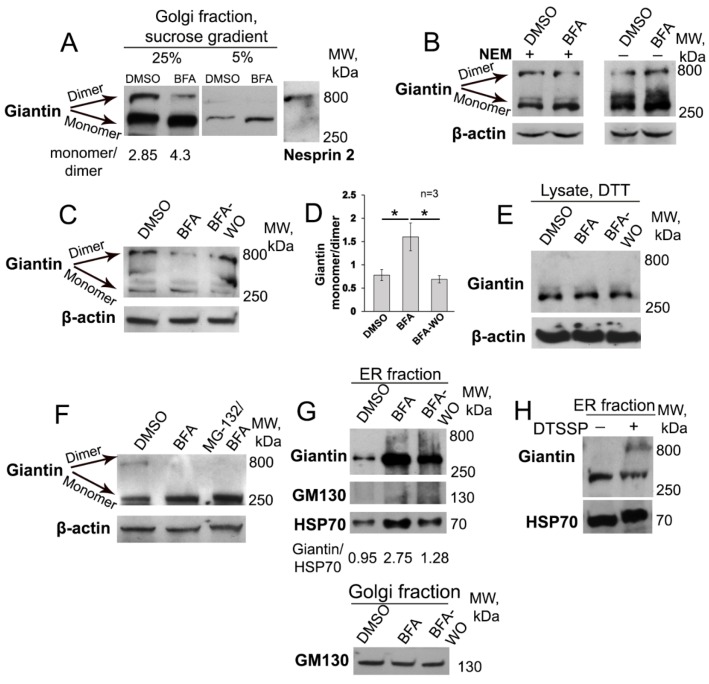
Giantin dimerization in Brefeldin A (BFA) and BFA-washout (WO) cells. (**A**) HeLa cells were treated with 36 μM BFA for 1 h or with a corresponding amount of DMSO (control). The Golgi fractions isolated from cells were subjected to sucrose sedimentation analysis on a 5–25% sucrose gradient. The 25% and 5% fractions were collected and analyzed by 4–15% SDS-PAGE. The samples were prepared under low (1%) concentrations of β-mercaptoethanol, and the same amount of proteins were loaded. Giantin-dimer and monomer are indicated by arrows. The size marker indicates 800 kDa protein Nesprin, which corresponds to the giantin dimer. Densitometry analysis of ratio monomer/dimer is presented below. (**B**) Giantin W-B of the lysates of HeLa cells treated with DMSO and BFA for 1 h. The lysates were prepared in the presence (+) or absence (−) of 2 mM NEM followed by 1% β-mercaptoethanol and resolved by 8% SDS-PAGE. (**C**) Giantin W-B of the lysates of HeLa cells treated with DMSO, BFA for 1 h, and BFA-WO for another 60 min. The lysates were prepared under 2 mM NEM followed by 5% β-mercaptoethanol and resolved by 10% SDS-PAGE. Bottom panel: β-actin as a loading control. (**D**) Quantification of the band’s intensity monomer/dimer from three independent experiments presented in (**C**). Data are presented as a mean ± SD; * *p* < 0.001. (**E**) Giantin W-B of the lysates of HeLa cells: DMSO, BFA, and BFA-WO. Samples were prepared under 2 mM NEM followed by 20 mM DTT and run on 10% SDS-PAGE. (**F**) Giantin W-B of the lysates of HeLa cells: DMSO, BFA, and treated with 5 µM MG-132 for 3 h followed by BFA. Samples were prepared under 2 mM NEM followed by 10% β-mercaptoethanol and run on 10% SDS-PAGE; β-actin was a loading control. The data are representative from three independent experiments. (**G**) Top panel: Giantin and GM130 W-B of the ER (microsomal) fraction isolated from HeLa cells: DMSO, BFA, and BFA-WO for 30 min. Samples were prepared under 2 mM NEM followed by 10% β-mercaptoethanol and run on 4–15% SDS-PAGE; HSP70 was a loading control. Densitometry analysis of ratio giantin/HSP70 is presented below. The data are representative from three independent experiments. Bottom panel: GM130 W-B of the Golgi fractions isolated from the cells presented in the top panel. (**H**) Giantin W-B of the microsomal fractions of HeLa cells incubated in presence of 2 mM DTSSP at RT. Samples were prepared under non-reducing conditions to avoid cleavage of thiol groups and run on 4–15% SDS-PAGE; HSP70 was a loading control.

**Figure 2 cells-08-01631-f002:**
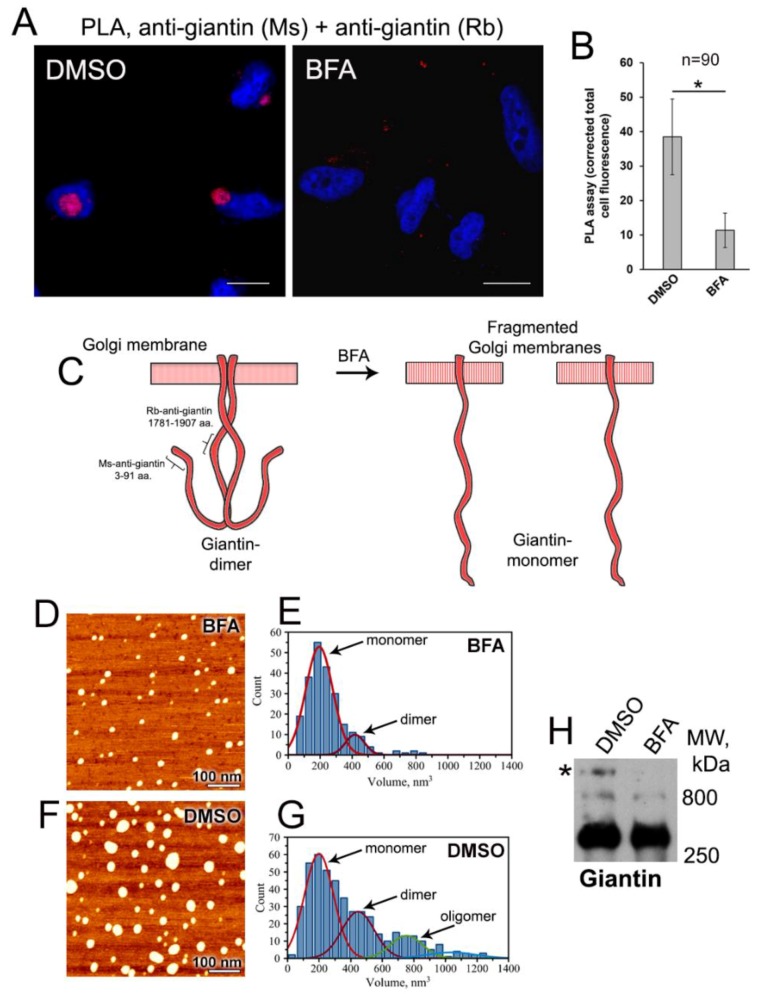
Proximity ligation assay (PLA) of giantin in HeLa cells. (**A**) Cells were treated with DMSO or BFA for 1 h. The proximity of giantin’s N-terminus and the central region of its cytoplasmic domain was evaluated using mouse anti-giantin 3–91aa and rabbit anti-giantin 1781–1907aa Abs. Red punctae indicate PLA signal, nucleus is in blue, DAPI; bars, 10 μm. (**B**) Quantitation of proximity ligation is presented as the corrected total fluorescence intensity (a.u.). The results are measured as a mean ± SD; *n* = 90 cells from three independent experiments; * *p* < 0.001. (**C**) Schema illustrating a possible conformation of giantin dimer in control and BFA-treated cells; the epitopes of Abs are indicated by brackets. Presumably, the cytoplasmic domain of giantin dimer is flexed in a non-coiled coil domain allowing closeness of N-terminus and the central region. BFA-induced de-dimerization results in the loss of PLA signal, presumably over the extension of the cytoplasmic domain. (**D**) AFM topography image of giantin from BFA-treated cells. (**E**) Statistical histogram of protein volume distribution for BFA-treated sample (*n* = 231). (**F**) AFM topography image of giantin from DMSO-treated cells. (**G**) Statistical histogram of protein volume distribution for DMSO-treated sample (*n* = 457). XY-scale is as indicated by the white bar = 100 nm, Z-scale is 1.6 nm. (**H**) Giantin W-B of the lysates of HeLa cells treated with DMSO and BFA for 1 h. The lysates were prepared without reducing agents and resolved by 4–15% SDS-PAGE. An asterisk (*) indicates the band corresponded to the oligomeric giantin.

**Figure 3 cells-08-01631-f003:**
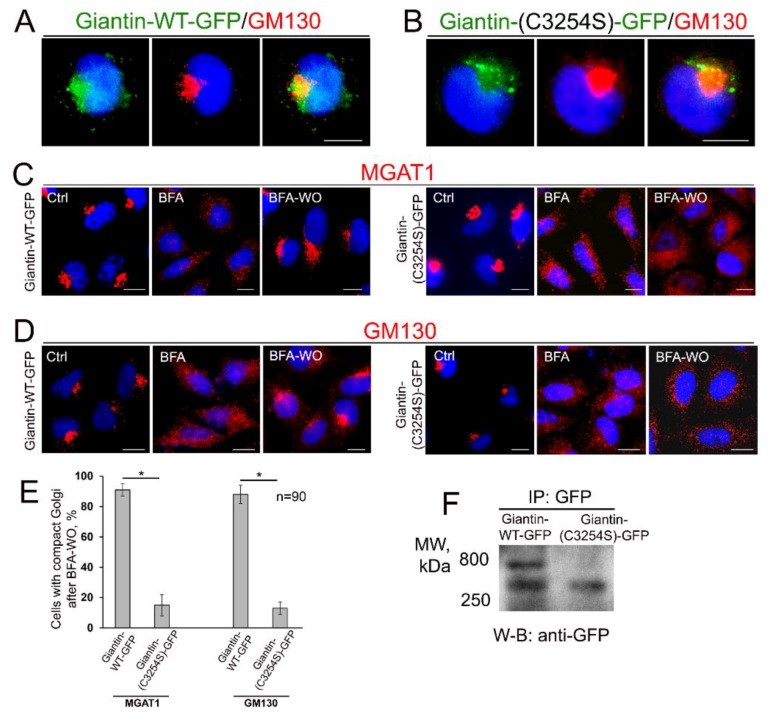
The disulfide bond in the luminal domain of giantin is critical for its dimerization and Golgi biogenesis. (**A**,**B**) Confocal immunofluorescence images of Golgi in HeLa cells transfected with WT giantin tagged with GFP at the C-terminus or with the same construct mutated at Cys3254 to Ser. Cells were stained with GM130 to validate localization in Golgi. (**C**,**D**) Immunostaining of MGAT1 (**C**) or GM130 (**D**) in HeLa cells overexpressing giantin-WT-GFP or giantin(-C3254S)-GFP and treated with 36 μM BFA for 60 min and then WO for 60 min. Control cells were exposed to the corresponding amount of DMSO. Images were captured using the EVOS M5000 Imaging System (Thermo Fisher Scientific, Waltham, MA, USA). Nuclei counterstained with DAPI (blue); bars, 10 μm. (**E**) Quantification of cells with perinuclear Golgi after BFA-WO for indicated Golgi markers; *n* = 90 cells from three independent experiments. Results are expressed as a mean ± SD; * *p* < 0.001. (**F**) GFP W-B of the GFP IP isolated from the lysates of HeLa cells transfected with giantin-WT-GFP or giantin(-C3254S)-GFP. The lysates were prepared in the presence or absence of 2 mM NEM followed by 5% β-mercaptoethanol and resolved by 4–15% SDS-PAGE. Samples were normalized by the total protein concentration.

**Figure 4 cells-08-01631-f004:**
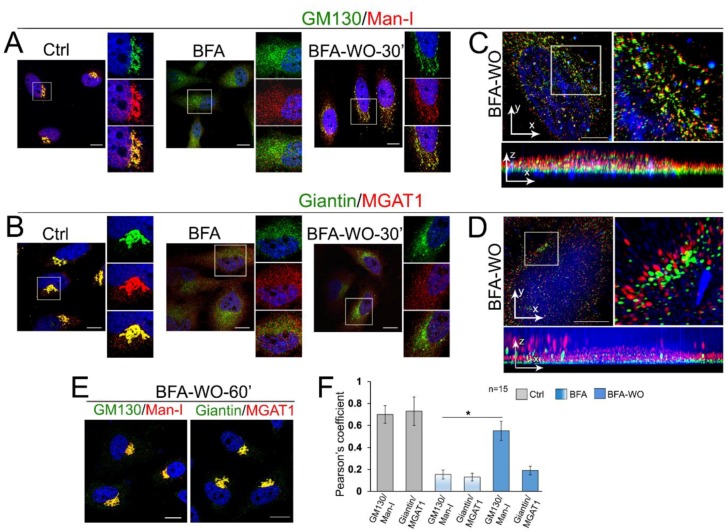
The differential Golgi targeting mechanism for enzymes during Golgi biogenesis. (**A**,**B**) Immunostaining of GM130 and Man-I (**A**), and giantin and MGAT1 (**B**) in HeLa cells: control (DMSO-treated), BFA-treated, and BFA-WO for 30 min. White boxes indicate Golgi areas enlarged and shown in three channels on the right side. Nuclei were counterstained with DAPI (blue). All confocal images were acquired with the same imaging parameters; bars, 10 μm. (**C**,**D**) Representative 3D SIM imaging of HeLa cells after BFA-WO. Cells were co-stained with GM130 and Man-I (**C**), and giantin and MGAT1 (**D**). The Golgi area in the white boxes is enlarged and presented on the right side. The orthogonal views of each area are shown below; bars, 10 μm. (**E**) Immunostaining of GM130 and Man-I, and giantin and MGAT1 in HeLa cells recovered after BFA for 60 min. (**F**) Quantification of Pearson’s coefficient of colocalization for indicated proteins in cells presented in C and D; *n* = 15 cells for each series of SD SIM imaging; results expressed as a mean ± SD; * *p* < 0.001.

**Figure 5 cells-08-01631-f005:**
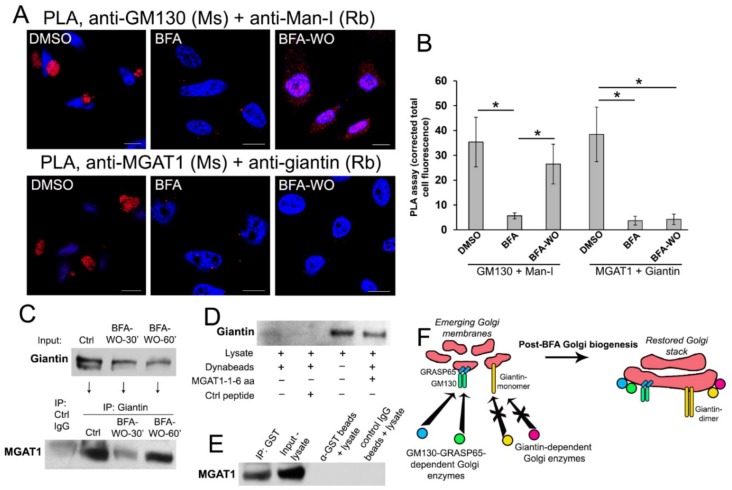
Interaction of MGAT1 and giantin. (**A**) PLA in HeLa cells treated with DMSO, BFA, and BFA-WO. The proximity of MGAT1 and giantin was evaluated using mouse anti-MGAT1 and rabbit anti-giantin Abs. The closeness of GM130 and Man-I was examined using mouse anti-GM130 and rabbit anti-Man-I Abs. Red punctae indicate PLA signal, nucleus is in blue, DAPI; bars, 10 μm. (**B**) Quantitation of proximity ligation for indicated proteins is presented as the corrected total fluorescence intensity (a.u.). The results are measured as a mean ± SD; * *p* < 0.001. (**C**) Bottom panel: MGAT1 W-B of the complexes pulled down by anti-giantin Ab from the Golgi fractions isolated from HeLa cells: control, and recovered at 30 and 60 min of BFA-WO. The input of giantin is presented at the top panel. The IP using control rabbit IgG served as a control. (**D**) Giantin W-B of the lysate or the complex pulled down from the lysates of non-treated HeLa cells using biotinylated hMGAT1 N-terminal peptide and Dynabeads M-280 Streptavidin. Control samples are the Dynabeads incubated with cell lysate in the presence (+) or absence (−) of control peptide. (**E**) MGAT1 W-B of the complexes pulled down from the lysate of non-treated HeLa cells using giantin-GST N-terminal peptide immobilized by anti-GST Ab coupled epoxy beads. The anti-GST beads exposed to the cell lysate only and IP using control rabbit IgG are the control. (**F**) The proposed model of Golgi targeting. Enzymes that employ GM130-GRASP65 docking site are able to reach membranes during Golgi biogenesis, contrary to other Golgi resident proteins that use giantin. The targeting of the latter occurs after giantin dimerization and complete restoration of Golgi morphology.

**Figure 6 cells-08-01631-f006:**
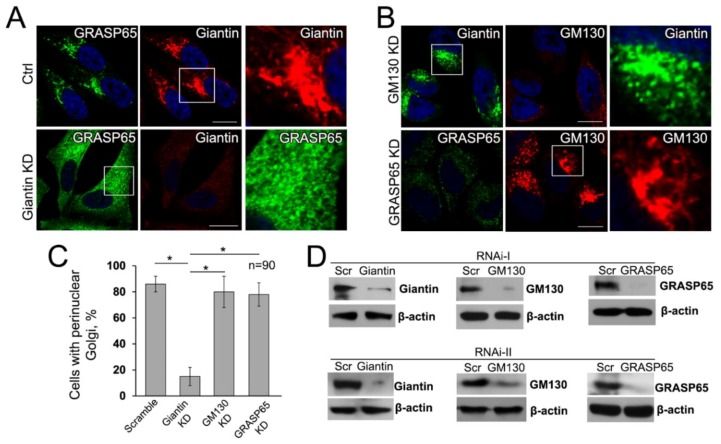
Giantin is necessary for the restoration of compact Golgi upon BFA washout. (**A**,**B**) Confocal immunofluorescence images of Golgi were collected in HeLa cells pretreated with different siRNAs for 72 h followed by exposure to 36 μM BFA for 60 min and then WO for 30 min. The combination of GRASP65 + giantin immunostaining was used for cells treated with scramble or giantin siRNAs, and giantin + GM130 and GRASP65 + GM130 were used for cells treated with GM130 and GRASP65 siRNAs, respectively. Images in the white boxes are enlarged and displayed as either green or red channels on the right side. Nuclei counterstained with DAPI (blue); bars, 10 μm. (**C**) Quantification of cells with perinuclear Golgi, as shown in (**A**,**B**); *n* = 90 cells from three independent experiments of giantin, GM130, and GRASP65 KD performed with two different combinations of target-specific siRNAs. Results are expressed as a mean ± SD; * *p* < 0.001. (**D**) Giantin, GM130, and GRASP65 W-B of lysates of HeLa cells treated with the corresponding siRNAs (I and II); β-actin was a loading control.

**Figure 7 cells-08-01631-f007:**
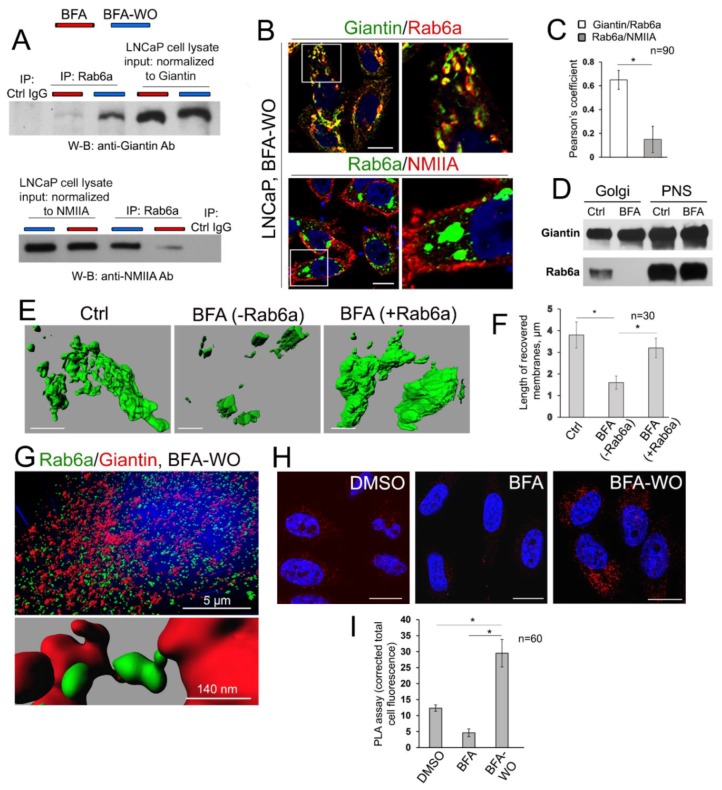
Rab6a-mediated Golgi biogenesis after BFA-WO. (**A**) Giantin (top panel) and NMIIA (bottom panel) W-B of complexes pulled down with anti-Rab6a Ab from lysates of LNCaP cells treated with BFA for 60 min and then 30 min WO. Amounts of lysates used for IP were normalized to giantin or NMIIA, as described in Section 2. Non-specific rabbit IgG was used for control IP. Red and blue markers indicate BFA and BFA-WO samples, respectively. (**B**) Confocal immunofluorescence images of Rab6a colocalization with giantin and NMIIA in LNCaP BFA-WO cells. (**C**) Quantification of Pearson’s overlap coefficient for indicated pairs of stained proteins in cells after BFA-WO. All confocal images were acquired with the same imaging parameters; bars, 10 μm. Data collected from 90 cells of three independent experiments, results are expressed as a mean ± SD; * *p* < 0.001. (**D**) Giantin and Rab6a W-B of the postnuclear supernatant (PNS) and Golgi membranes collected from the HeLa cells: control and treated with BFA. (**E**) The representative 3D reconstruction of the SIM imaging of Golgi membranes isolated from HeLa cells: non-treated, and treated with BFA followed by incubation with or without Rab6a protein. The regenerated Golgi membranes were stained with giantin; bars, 2 µm. (**F**) Quantification of the average length of Golgi membranes (*n* = 30 from each sample) presented in (**E**). (**G**) Top panel: the reconstructed 3D SIM imaging of Golgi in HeLa cells after 30 min of BFA-WO. Cells were stained with Rab6a (green) and giantin (red). The representative area of Golgi is presented in the bottom panel. Note the Rab6a punctae between giantin-positive emerging Golgi membranes. (**H**) PLA in HeLa cells treated with DMSO, BFA, and BFA-WO. The proximity of giantin and Rab6a was evaluated using mouse anti-giantin and rabbit anti-Rab6a Abs. Red punctae indicate PLA signal, nucleus is in blue, DAPI; bars, 10 μm. (**I**) Quantitation of proximity ligation for Rab6a and giantin is presented as the corrected total fluorescence intensity (a.u.). The results are measured as a mean ± SD; * *p* < 0.001.

**Figure 8 cells-08-01631-f008:**
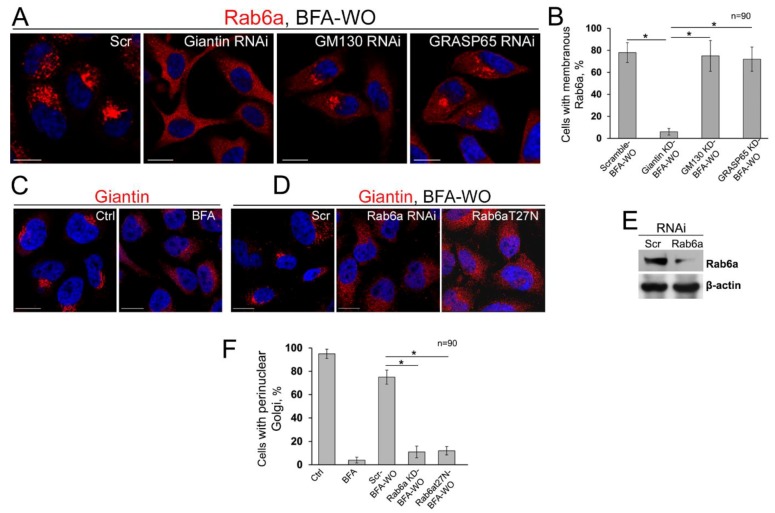
The overlap of giantin and Rab6a during Golgi biogenesis. (**A**) Confocal immunofluorescence images of Rab6a in HeLa cells after 60 min of BFA-WO, pretreated with scramble, giantin, GM130, or GRASP65 siRNAs. All confocal images acquired with the same imaging parameters; bars, 10 μm. (**B**) Quantification of cells with membranous Rab6a in cells presented in (**A**); *n* = 90 cells from three independent experiments, results are expressed as a mean ± SD; * *p* < 0.001. (**C**) Giantin immunostaining in DMSO- and BFA-treated HeLa cells. (**D**) Giantin immunostaining in HeLa cells after 60 min of BFA-WO, transfected with scramble, Rab6a siRNAs, and dominant-negative (GDP-bound) Rab6a(T27N). (**E**) Rab6a W-B of lysates of HeLa cells treated with corresponding siRNAs; β-actin was a loading control. (**F**) Quantifications of cells with perinuclear Golgi in cells from (**C**,**D**); *n* = 90 cells from three independent experiments, results expressed as a mean ± SD; *, *p* < 0.001.

**Figure 9 cells-08-01631-f009:**
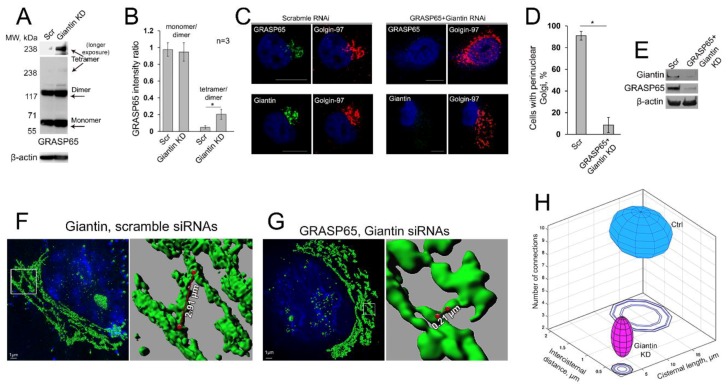
Differential Golgi phenotype in control and giantin-depleted cells. (**A**) GRASP65 W-B of the lysate of HeLa cells: treated with a scramble and giantin siRNAs; β-actin was a loading control. Longer exposure for GRASP65-tetramer is presented in the top panel. (**B**) Quantification of the intensity of bands corresponding to GRASP65: monomer/dimer and tetramer/dimer. Calculations performed within the same exposure, and data represent mean ± SD from three independent experiments; * *p* < 0.001. (**C**) Confocal immunofluorescence images of GRASP65 and Golgin-97 or giantin and Golgin-97 in HeLa cells treated with scramble or a mix of GRASP65 and giantin siRNAs; bars, 10 μm. (**D**) Quantifications of cells with perinuclear Golgi in cells from C; *n* = 90 cells from three independent experiments, results expressed as a mean ± SD; * *p* < 0.001. (**E**) Giantin and GRASP65 W-B of the HeLa cells treated with scramble or mix of GRASP65 and giantin siRNAs; β-actin was a loading control. (**F**,**G**) 3D SIM imaging of giantin and GRASP65 in HeLa cells transfected with a scramble and giantin siRNAs, respectively; bars, 1 μm. Red lines indicate the size of intercisternal connections. The representative area of intercisternal connections is highlighted by white boxes and presented on the right side. (**H**) Statistical analysis of cisternal length, size of intercisternal connections, and their number. Clear separation of observations in control and giantin KD HeLa cells visualized in an XYZ plot where mean and SD were calculated for each parameter (axis) and then used to display an ellipsoid for both conditions. The large ellipsoid represents control data, the small ellipsoid represents giantin KD. Contour (shadow) of ellipsoids presented at Z = 0. Hypothesis testing for different medians was performed via non-parametric Wilcoxon rank-sum test. *p*-values for length of cisternae, intercisternal distance, and number of connections are 2.2544 × 10^−29^, 1.2278 × 10^−16^, 5.2380 × 10^−4^, respectively, with Wilcoxon.

**Figure 10 cells-08-01631-f010:**
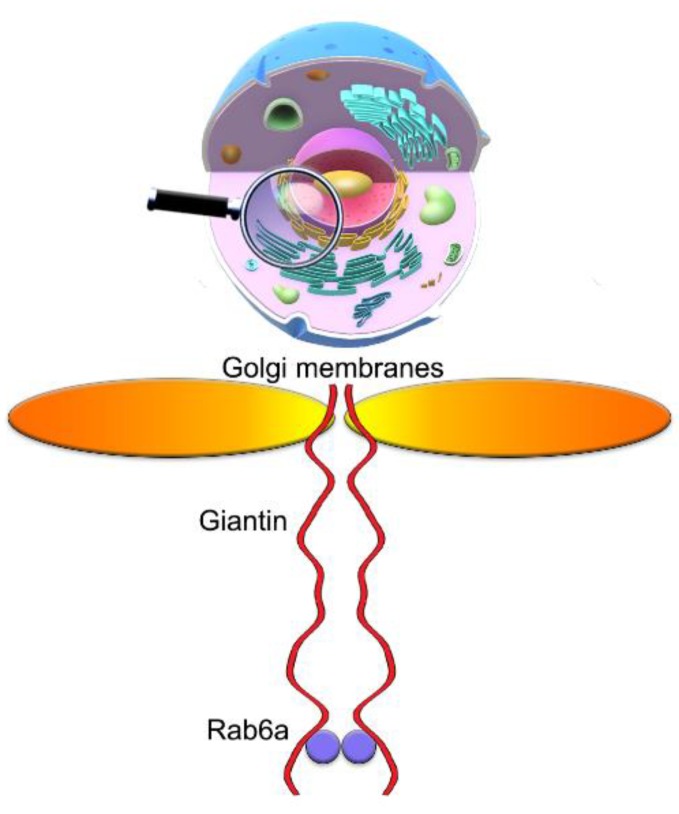
Schematic illustrations of Golgi biogenesis. The fusion of Golgi membranes (highlighted below) is presumably initiated by tethering of giantin from the rim of one cisterna to the rim of opposite cisterna via disulfide bond in the luminal domain. Interaction of gaintin and Rab6a via their N-terminus may provide twisting of giantin monomers required for the coiled-coil dimeric structure. Such fusion requires the force presumably created by the action of F-actin based motor protein, non-muscle Myosin IIB.

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
