# Peer review of "Post-ER Stress Biogenesis of Golgi Is Governed by Giantin"

_cells, 2019, doi:10.3390/cells8121631_

Round 1
Reviewer 1 Report
The authors investigate the effects of BFA on Giantin, a Golgi matrix protein, and its involvement in restoration of Golgi morphology following BFA treatment. BFA-induced Golgi dissolution is associated with Giantin monomer formation, whereas Giantin dimerization coincides with fusion of Golgi membranes and Golgi complex reestablishment following BFA washout which furthermore depends on Rab6a. The authors use a number of cell biological, biochemical and microscopic approaches that together are well supporting the main conclusions of this paper.
Specific comments:
The authors should use also Golgi-disrupting agents other than BFA (e.g. DNA damage inducers, HDAC inhibitors, GCA, M-COPA) to learn more about the generalizability of their data: is this a BFA-specific effect only? In this regard, are there physiological settings in which Giantin dimerization/Golgi structure is affected and thus mimic the effects of BFA? Dispersal of the Golgi in different cancer cell lines can be achieved with nanomolar concentrations. Throughout the paper in their experiments the authors use an extremely high BFA concentration (36 uM). BFA can also lead to ADP-ribosylation of other cellular factors impacting their function (e.g BARS). I would like to see whether some of the authors’ results (e.g. Giantin dimerization) can be recapitulated with much lower BFA doses. Figure S6: Knockdown validation (Giantin, GRASP65, GM130) is missing Giantin knockdown prevents restoration of normal Golgi morphology following BFA washout – does enforced Giantin expression protect from BFA-induced Golgi dispersal? Conceptually, it would make sense to present the data/experiments describing the effect of the giantin[C3254S] mutant on dimerization together with the PLA data from Figure 2 (currently presented as Fig. 6)Author Response
Response to the Reviewers’ comments
First of all, we appreciate both Reviewers for their positive appraisal of our story and would like to thank them for reviewing this manuscript and providing detailed comments. We have addressed the concerns of Reviewers and revised the manuscript according to their recommendations.
Reviewer #1:
Q-1: The authors should use also Golgi-disrupting agents other than BFA (e.g. DNA damage inducers, HDAC inhibitors, GCA, M-COPA) to learn more about the generalizability of their data: is this a BFA-specific effect only? In this regard, are there physiological settings in which Giantin dimerization/Golgi structure is affected and thus mimic the effects of BFA?
A-1: We appreciate this comment, as this aspect has an important general application. Indeed, we previously showed that alcohol-induced Golgi fragmentation is associated with giantin de-dimerization (Petrosyan et al. Sci Rep. 2015;5:17127. PMID: 26607390). Moreover, we found that under normal conditions in low-aggressive androgen-sensitive prostate cancer cells, LNCaP, Golgi is compact and perinuclear, and giantin dimerization is not affected, however, in both PC-3 and DU145 cells, which represent aggressive androgen-restrictive prostate cancer cells, Golgi is fragmented and giantin is monomerized (Petrosyan et al. Mol Cancer Res. 2014 12(12):1704-16. PMID: 25086069). Intriguingly, the expression of ER-stress specific proteins in PC-3 and DU145 cells is higher than that in LNCaP cells. Generally speaking, ER-stress and giantin de-dimerization are linked to each other. Also, we recently found that LNCaP cells treated with ER-stress inducer Thapsigargin demonstrate fragmented Golgi phenotype and monomerization of giantin (see below). We did not include these data in the current story (we like to present these data in the separate manuscript) but mentioned this aspect in the updated version of the manuscript.
Q-2: Throughout the paper in their experiments the authors use an extremely high BFA concentration (36 uM). BFA can also lead to ADP-ribosylation of other cellular factors impacting their function (e.g BARS). I would like to see whether some of the authors’ results (e.g. Giantin dimerization) can be recapitulated with much lower BFA doses.
A-2: The dosage of BFA at 36 µM is widely used to study the disorganization of Golgi. Some authors (Jesch and Linstedt. Mol Biol Cell. 1998 Mar;9(3):623-35) even used long exposure of this concentration of BFA than we did in the current story. In the manuscript “Molecular mechanism and functional role of brefeldin A-mediated ADP-ribosylation of CtBP1/BARS” by Colanzi et al. (PNAS, 2013 110 (24) 9794-9799) authors use 285 µM BFA. However, we appreciate this helpful comment and have performed a series of analyses to address it. We kept cells under 18 µM of BFA for 2 hours, and Golgi disorganization was associated with giantin monomerization (Fig. S1B and C, see below). Next, we reduced BFA concentration to 10 times less than we used but kept treatment for 10 hours. Again, Golgi was disorganized and giantin was de-dimerized (Fig. S1D and E, see below). At the concentration of 720 nM, we have not noticed the effect of BFA on the Golgi (Fig. S1F, see below). These results are included in the current Fig. S1, and the appropriate changes are made in the text of manuscript.
Q-3: Figure S6: Knockdown validation (Giantin, GRASP65, GM130) is missing Giantin knockdown prevents restoration of normal Golgi morphology following BFA washout.
A-3: We did not include the W-B data because KD validation was visible on the immunofluorescence pictures. Now, in the updated Fig. S7, we presented all W-B results regarding KD of indicated proteins.
Q-4: Does enforced Giantin expression protect from BFA-induced Golgi dispersal?
A-4: In respect to response to BFA, we have not seen the difference between HeLa cells with WT giantin and HeLa cells overexpressing giantin-GFP. We believe that effect of BFA is rather mediated by the interaction of motor protein non-muscle Myosin IIA with resident Golgi enzymes but not with golgins like giantin. Details can be found in: 1) Petrosyan and Cheng. Glycobiology. 2013 Jun;23(6):690-708). PMID: 23396488; 2) Duran et al. Mol Biol Cell. 2003 Feb;14(2):445-59. PMID: 8082645.
Q-5: Conceptually, it would make sense to present the data/experiments describing the effect of the giantin[C3254S] mutant on dimerization together with the PLA data from Figure 2 (currently presented as Fig. 6).
A-5: We appreciate this very helpful comment and moved Figure 6 right next to Fig. 2. Now these data presented as Fig. 3.

Reviewer 2 Report
In general, there are many WB that lack loading controls or seem to have sub-optimal exposure.
Specific remarks:
Movie S1 – scale bar . No quantification of apoptosis or statistics.
Movie S2 – Why does the nuclear signal decrease with time? Could be a result of photobleaching? If so, this can explain the decrease in GFP as well. The quantification for the process is also missing.
Fig 1 – actin loading control is missing in some of the figures, as well as the Mw ladder for protein size.
1B – why is there no difference between DMSO and BFA?
1D – How was this quantified? How many repeats were done?
1E – a positive control for the dimer is missing.
1F – lack of quantification and repeats.
1G – looks like uneven loading of the gel, and a positive control for GM130 is missing.
Page 4 – lines 163 – line break is wrong.
Figure 2A,B – how many cells were analyzed?
2C,D – there is no demonstration that GFP-Giantin behaves as the native protein, without GFP.
Figure 5A,B – the choice of panels is not clear and/or consistent.
Figure 6E – How was the quantification done?
In materials and methods, the description of microscopy methods is not complete (as lasers, filters, exposure etc.)
Author Response
Response to the Reviewers’ comments
First of all, we appreciate both Reviewers for their positive appraisal of our story and would like to thank them for reviewing this manuscript and providing detailed comments. We have addressed the concerns of Reviewers and revised the manuscript according to their recommendations.
Reviewer #2:
Comments and Suggestions for Authors
Q-1: In general, there are many WB that lack loading controls or seem to have sub-optimal exposure.
A-1: We added loading controls to some of the figures. Also, we would like to note that some W-B pictures require low exposure to avoid the fusion of oversaturated adjacent bands related to giantin monomer. This is critical for proper quantification of ratio between monomer- and dimer-specific giantin bands.
Specific remarks:
Q-2: Movie S1 – scale bar. No quantification of apoptosis or statistics.
A-2: We replaced this movie file with a new video combining three movies of one HeLa cell treated with BFA for 3 h. The reason to stop video capturing was (a) to adjust laser intensity in order to keep the same intensity of GFP, which is normally bleached within 1 h, and (b) refocus the cell, as it is moving out from the vision area over motility. So, in the new movie, scale bar and timing are included. Also, in the updated Fig. S1 we included the level of caspase-3 from the cells treated with BFA for 3 h. Additionally, we made the changes in the manuscript’s text: “We tested this possibility in HeLa cells treated with 36 μM BFA for 1 hour, only because prolonged (up to 3 hours) treatment with BFA is known to launch the apoptosis, as indicated by the enhanced expression of caspase-3 (Fig. S1A), membrane blebbing, and chromatin condensation (Movie S1)”
Q-3: Movie S2 – Why does the nuclear signal decrease with time? Could be a result of photobleaching? If so, this can explain the decrease in GFP as well. The quantification for the process is also missing.
A-3: We appreciate this comment from the Reviewer. Yes, we noticed that GFP, as wells as a nuclear Hoechst-specific nuclear staining are bleached after 35-40 min. The use of DMEMgfp-2 medium (Axxora, NY, USA) improved life visualization. In the updated Movie S2, the effect of BFA on Golgi precedes major changes in the intensity of green or blue color. Also, the timing and scale bars are included.
Q-4: Fig 1 – actin loading control is missing in some of the figures, as well as the Mw ladder for protein size.
A-4: We included the beta-actin loading control only for the lysate samples. Please note that for sucrose sedimentation analysis, samples were normalized by the total protein concentration (it was indicated in the figure legends). The use of any loading control for this experiment can be misleading. Also, for the ER fractions samples, we used HSP70, which is more sensitive and reliable loading control than beta-actin. Also, MW marker is denoted for all figures.
Q-5: 1B – why is there no difference between DMSO and BFA?
A-5: The purpose of the experiments presented in Fig. 1S is to show that the use of N-ethylmaleimide (NEM), which blocks free sulfhydryls, prevents the self-dimerization of giantin-monomers during the lysis of cells. In the left panel, in the presence of NEM, the dimer-specific band was reduced, how in absence of NEM, monomers can form dimer, thus compromising the effect of BFA. In our further experiments, all BFA-treated samples were prepared in presence of NEM.
Q-6: 1D – How was this quantified? How many repeats were done?
A-6: Densitometric analysis of band intensity was performed using ImageJ. It was indicated in the Methods section – Miscellaneous. The number of repeats is three (n=3), it was denoted in the right corner of the diagram. Also, this statistic was recalculated.
Q-7: 1E – a positive control for the dimer is missing.
A-7: The positive control for this experiment (giantin-dimer) is presented in Fig. B and C, when cells were treated with M-E but not DTT. We believe that there is no need to present it again. The actual control for this experiment is the DMSO-treated cells.
Q-8: 1F – lack of quantification and repeats.
A-8: We would like to note that in these series of experiments we did not notice prominent giantin-dimer band in BFA-treated or MG-132/BFA treated samples. This was only one reason not to present the quantification dimer/monomer. The data provided in Fig. 1F is representative of three independent experiments. The data from the other two experiments are presented below.
Q-9: 1G – looks like uneven loading of the gel, and a positive control for GM130 is missing.
A-9: We agree that the BFA-treated sample is overloaded according to the HSP70 signal. However, the densitometric quantification of giantin/HSP70 is presented below, which clearly indicates the increase of giantin in the ER after BFA. Also, please note that GM130 is normally not presented in the ER. Notably, after BFA treatment, giantin moves to the ER while GM130 still retains in the fragmented Golgi membranes. This aspect has repeatedly shown by different groups and was highlighted in the Results and Discussion parts of our manuscript. Thus, we presented a GM130-specific probe to demonstrate that ER fraction is not contaminated by Golgi membranes. In the updated figure, we show Golgi fractions from the same cells to demonstrate the positive control for GM130. Below, are the two other repeats of the same experiment.
Q-10: Page 4 – lines 163 – line break is wrong.
A-10: Thanks, it has been fixed.
Q-11: Figure 2A,B – how many cells were analyzed?
A-11: We would like to thank the Reviewer for this comment. It is n=90 cells from three independent experiments. The appropriate changes are made in the Figure legend.
Q-12: 2C,D – there is no demonstration that GFP-Giantin behaves as the native protein, without GFP.
A-12: We presented in the Fig. S4A pictures of response of HeLa cells expressing giantin-GFP to treatment with BFA. In control cells, giantin-GFP localizes in the perinuclear space, but redistributes to the cytoplasm after BFA treatment and returns to the Golgi after BFA-WO.
Q-13: Figure 5A,B – the choice of panels is not clear and/or consistent.
A-13: We made some rearrangements in the former Fig. 5. Now it is a Fig.6. I hope, it will facilitate readers’ understanding.
Q-14: Figure 6E – How was the quantification done?
A-14: We would like to clarify the determination of compact and fragmented Golgi, which we used previously (Petrosyan and Cheng. Glycobiology. 2013 Jun;23(6):690-708). PMID: 23396488). The whole-cell is divided into three areas: nuclear (A), perinuclear (B), and cytoplasmic (C), as schematized by three concentric ellipsoids. After subtraction of the nuclear region, the cytoplasm is considered as two-third of B+C, while one-third of this sum is denoted as perinuclear area, where the Golgi is located. We consider the Golgi fragmented if separate Golgi membranes are detected beyond the perinuclear area. Otherwise, we denote cells as having compact Golgi.
Please note that as recommended by Reviewer#1, we moved Fig. 6 right next to Fig. 2, and in the updated version of the manuscript this figure is denoted as Fig. 3.
Q-15: In materials and methods, the description of microscopy methods is not complete (as lasers, filters, exposure etc.).
A-15: Information is added to the Methods section.
